# Too old to telework? Age but not gender shapes hiring biases across telework and office settings

Kyriaki Fousiani[ID]1*, Sylvia Xu[ID]2¤, Bibiana M. Armenta Gutierrez3, Chloe Sypes[ID]4

1 Faculty of Behavioural and Social Sciences, University of Groningen, Groningen, The Netherlands, 2 Faculty of Economics and Business, University of Groningen, Groningen, The Netherlands, 3 Faculty of Behavioural and Movement Sciences, Free University Amsterdam, Amsterdam, The Netherlands, 4 Independent Researcher, Philadelphia, Pennsylvania, United States of America

¤ Current Address: Faculty of Psychology and Neuroscience, Maastricht University, Maastricht, The Netherlands
* k.fousiani@rug.nl

## Abstract

As teleworking becomes increasingly common in the post-pandemic workplace, recruiters may rely more on readily visible applicant characteristics (e.g., age and gender) when evaluating candidates, potentially triggering stereotype-based biases. This research investigates how work setting (telework versus in-office) interacts with applicant demographics to shape hiring recommendations. Drawing on Stereotype Content Theory, the study also examines whether perceptions of applicant warmth and competence mediate these effects. Across three vignette-based experimental studies, participants assessed applicants for positions offered in telework versus in-office settings. Studies 1 and 2 manipulated applicant age (younger versus older) and gender (female versus male), revealing a consistent preference for older applicants in in-office over telework positions, with no significant gender effects. Building on these findings, Study 3 focused solely on applicant age, employing a more nuanced three-level age design (younger, middle-aged, older) and replicated the age-by-setting effect. This preference for older applicants in office-based positions was mediated by perceptions of warmth. These findings suggest that hiring decisions are shaped not just by applicant qualifications, but also by perceived fit between applicant (age-related) demographics and contextual demands of the job. Theoretical and practical implications for addressing age- and gender-related biases in modern work contexts are discussed.

## Introduction

Teleworking has become increasingly common in recent years, particularly following the Covid-19 pandemic, which significantly altered the structure and norms of workplace organization. Importantly, the criteria for employee selection appear to differ between teleworking and traditional office-based work [1]. This is because

**Data availability statement:** Open Science and Transparency Practices Both Study 1 (https://osf.io/a9utq/overview; DOI number: 10.17605/OSF.IO/A9UTQ) and Study 2 (https://osf.io/wq52z/overview; DOI number: 10.17605/OSF.IO/WQ52Z) were preregistered in Open Science Framework. The preregistrations include hypotheses concerning additional variables not addressed in this manuscript (e.g., goals). To maintain clarity and focus, we chose to exclude these from the main text. However, all related data are openly available on the Open Science Framework: https://doi.org/10.17605/OSF.IO/FQ2PE.

**Funding:** The author(s) received no specific funding for this work.

**Competing interests:** Authors declare that they have no conflict of interest.

teleworking tends to involve increased social isolation and fewer interpersonal interactions, such as those with colleagues, while in-office work positions involve closer interactions and ongoing face-to-face collaboration, informal communication, and real-time responsiveness within teams [2–5]. Given these differing demands, recruiters may rely on visible characteristics of job applicants, such as their demographic characteristics, as heuristics to infer how well the individual aligns with the expectations of teleworking versus office-based roles.

Considering that individuals' age and gender are among the most visible and socially salient identities [6,7], we argue that these characteristics may serve as key cues that influence recruiters' perceptions of applicant suitability across different work settings. Stereotypes associated with age and gender can shape assumptions about an applicant's competence, autonomy, and sociability—traits that are differentially valued in teleworking versus office-based contexts [1]. The present study therefore examines whether recruiters' preferences for job applicants, and their corresponding hiring recommendations, are shaped by how they perceive applicants' age and gender in relation to the work setting (i.e., whether the position involves teleworking or working from a company office). This focus is particularly important, as recruiters' perceptions can systematically advantage or disadvantage certain demographic groups depending on the work setting, potentially reinforcing bias and inequality in access to traditional in-office positions or flexible work arrangements. For instance, Derous and Decoster found that implicit age cues in résumés (e.g., older-sounding names or traditional extracurricular activities) led to lower hirability ratings for older applicants, indicating age-based discrimination even when explicit age information was absent [8]. Similarly, Erlandsson et al. demonstrated that gender could affect hiring decisions, with female applicants, particularly mothers, being rated differently compared to their male counterparts [9]. However, it remains unclear how such biases manifest across telework versus in-office positions and settings.

Biases against certain demographic groups can be meaningfully understood through the lens of Stereotype Content Theory [10,11], which posits that social perceptions are shaped along two primary dimensions: warmth and competence. According to this theory, older individuals are typically viewed as warm but less competent [11–17], while younger individuals tend to be seen as more competent but lacking in warmth [14,18]. Gender-related stereotypes follow a similar pattern: Men are commonly stereotyped as more agentic and competent than women [11,19,20], whereas women are often perceived as higher in warmth [19,21]. These stereotype-based perceptions may influence how recruiters evaluate applicants for teleworking versus office-based roles, depending on the traits they associate with success in each setting. Given that recruiters tend to place greater emphasis on applicant competence over warmth in teleworking contexts compared to office-based settings [1], we argue that a) younger applicants and b) male applicants, who are stereotypically perceived as more competent, are more likely to be recommended for hire in teleworking positions as compared to in office-based positions. Conversely, a) older applicants, and b) female applicants, who are often viewed as warmer and more socially adept, may be preferred for in-office positions (rather than teleworking), where interpersonal interaction and collaboration are more central.

The present research makes several key theoretical contributions to the literature on recruitment, teleworking, and workplace diversity. First, it extends prior work on selection processes by demonstrating that recruiters' hiring recommendations are not solely shaped by applicants' qualifications or work setting, but also by the interaction between age- and gender-based stereotypes and the specific requirements of teleworking versus office-based roles. In doing so, our study builds on and refines earlier findings (e.g., Fousiani et al., 1) by showing that the *perceived* fit between applicant demographic characteristics and the work setting plays a central role in shaping hiring preferences. Second, this research contributes to the broader literature on bias in personnel selection by emphasizing that even ostensibly neutral organizational changes, such as the shift to teleworking, can unintentionally activate and reinforce disparities in hiring and working decisions [22,23]. Finally, our research informs the Stereotype Content Theory [10,11] by showing how role demands, such as those in teleworking versus office-based settings, differentially activate the warmth and competence dimensions of social perception, which influences in turn, hiring intentions. Our study underscores the need for greater awareness of context-specific bias and calls for selection practices that account for how work settings interact with perceptions of applicants' demographic characteristics.

## Age, gender, and teleworking versus working from a company office: A stereotype content approach

The COVID-19 pandemic significantly accelerated the adoption of teleworking, with 12.3% of the European Union workforce working remotely in 2020 [24]. Demographic characteristics such as age and gender play a key role in shaping individuals' preferences for teleworking versus office-based work. Employees aged 24–54 are nearly twice as likely to telework as those aged 55–64 [25], likely due to their greater digital fluency [26] and prevailing stereotypes that portray older employees as less technologically competent [23,27]. Gender differences also persist: women are 25–32% less likely than men to telework [28], partly because they are overrepresented in non-portable roles [25] and men are more likely to occupy positions offering greater autonomy and bargaining power [29]. While prior research has largely focused on how age and gender shape employees' telework versus office preferences, this study investigates how these demographic characteristics influence *recruiters*' perceptions of job applicants and their hiring decisions in teleworking versus office-based contexts.

Drawing on Stereotype Content Theory [10,11,30], which proposes that social perceptions are structured along two core dimensions, namely warmth and competence, we examine how age and gender influence recruiters' impressions of candidate suitability for remote or in-office positions. *Warmth* reflects perceived intentions and encompasses traits such as friendliness, likeability, collaboration, and supportiveness, while competence captures perceived ability and includes traits such as intelligence, skills, agency, and efficacy. In the workplace in particular, these two dimensions may serve different organizational functions and are valued differently depending on the context [1,30–32]. Warmth-related traits (often described as a form of interpersonal "art"; [33]) are closely associated with superior emotional and social interactions, as well as with influence skills such as negotiation and persuasion [34]. These traits are essential for fostering collaborative relationships with colleagues and clients, promoting a supportive work climate, and contributing to long-term organizational health. In contrast, competence-related traits, such as analytical thinking, technical expertise, efficiency, and problem-solving ability, are primarily linked to task performance and goal achievement. These traits signal an individual's capacity to deliver results, make informed decisions, and contribute to organizational effectiveness.

People may place disproportionate emphasis on competence or warmth depending on the requirements of the work setting at hand. For example, Wojciszke and Abele [35] found that competence is valued more than warmth when it serves the perceiver's instrumental goals. Similarly, Fousiani et al. [31] found that in organizational contexts prioritizing economic/instrumental objectives, competence takes precedence over warmth in hiring decisions. Conversely, when organizational goals emphasize relational dynamics and social cohesion, warmth-related traits become more important. These findings underscore that while both warmth and competence are valued in the workplace, their relative importance in impression formation is context-dependent.

This context sensitivity is particularly relevant when comparing telework and office-based work settings, as each emphasizes different skill sets. As a result, recruiters may (unconsciously) amplify the importance of competence in remote work evaluations, making competence-related perceptions more salient than in traditional office settings [1] where warmth may be more highly valued. This shift in evaluative focus (competence or warmth) may interact with social stereotypes, especially those related to age and gender, potentially reinforcing biased perceptions of candidate suitability across different work settings.

Indeed, demographic characteristics like age and gender are particularly susceptible to strong and persistent stereotyping. For example, older employees are frequently stereotyped as warm but lacking in competence [12–17], whereas younger employees are generally seen as more competent but less warm [14,18]. Similarly, men are typically viewed as more agentic and competent, whereas women are often perceived as warmer but less competent [11,19–21]. These stereotype-driven perceptions may influence recruiters' evaluations differently across work settings. Further research supports the influence of these stereotypes on hiring decisions. For instance, older candidates may be disadvantaged due to perceptions of lower competence [36]. Additionally, perceptions of warmth and competence can vary depending on occupational stereotypes, with certain professions being associated with specific demographic groups. For example, women are more represented in occupations characterized by high warmth and low competence, aligning with traditional gender stereotypes [37].

### Current research and hypotheses

Building on this literature, the current study argues that work setting (telework versus office) will interact with the applicants' demographics (age and gender) in predicting recommendation for hire. This interaction will be explained by the differential warmth and competence levels attributed to the various demographic groups. Specifically, we propose that recruiters will be more likely to recommend a) older and b) female applicants for in-office compared to telework positions. This effect will be mediated by heightened perceptions of these applicants as warm, a trait deemed particularly valuable in interpersonal, face-to-face work environments. In contrast, a) younger and b) male applicants will be more likely to be recommended for telework positions compared to in-office positions. This relationship will be mediated by the higher perceived competence of these applicants, a trait considered crucial in socially sterile, remote contexts (see Figs 1 and 2 for an illustration of the conceptual model).

Based on the above, we stated the following hypotheses

### Work setting by applicant age and gender: The mediating role of warmth

*Hypothesis 1a* Recruiters will be more likely to recommend older applicants for in-office positions than for telework positions.

*Hypothesis 1b* The increased likelihood of recommending older applicants for in-office positions will be mediated by their perceived warmth. Specifically, older applicants will be viewed as warmer and therefore more suitable for recommendation in the in-office (than the telework) setting.

*Hypothesis 2a* Recruiters will be more likely to recommend female applicants for in-office positions than for telework positions.

*Hypothesis 2b* The increased likelihood of recommending female applicants for in-office positions will be mediated by their perceived warmth. Specifically, female applicants will be viewed as warmer and therefore more suitable for recommendation in the in-office (than the telework) setting.

### Work setting by applicant age and gender: The mediating role of competence

*Hypothesis 3a* Recruiters will be more likely to recommend younger applicants for telework than for in-office positions.

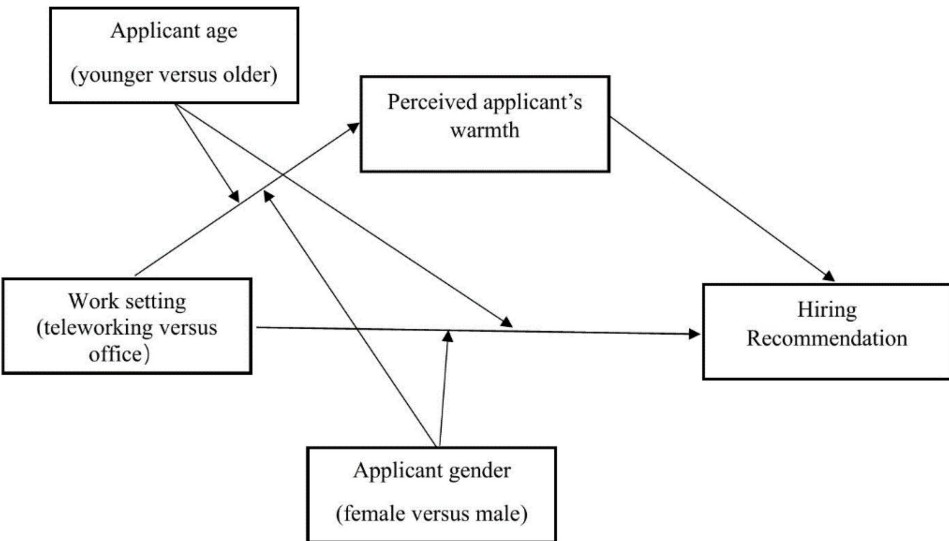

**Fig 1. Hypothesized theoretical model with perceived Applicant's Warmth as the mediator.**

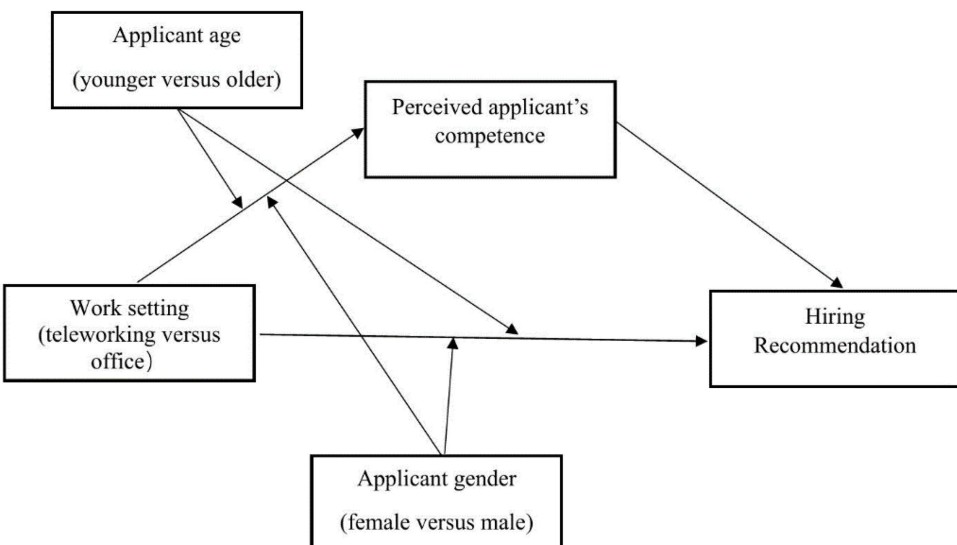

**Fig 2. Hypothesized theoretical model with perceived Applicant's competence as the mediator.**

*Hypothesis 3b* The increased likelihood of recommending younger applicants for telework positions will be mediated by their perceived competence. Specifically, younger applicants will be viewed as more competent and therefore more suitable for recommendation in the telework (than the office) setting.

*Hypothesis 4a* Recruiters will be more likely to recommend male applicants for telework than for in-office positions.

*Hypothesis 4b* The increased likelihood of recommending male applicants for telework positions will be mediated by their perceived competence. Specifically, male applicants will be viewed as more competent and therefore more suitable for recommendation in the telework (than the office) setting.

## Overview of the current research

We conducted three experimental studies to test our hypotheses. In all three studies, work setting (teleworking versus office) and applicant demographics were experimentally manipulated through vignettes, while the manipulation checks and dependent variables were subsequently measured. In each study, participants were presented with a scenario describing a job vacancy and were instructed to adopt the perspective of a recruiter evaluating one of the (supposedly) several shortlisted candidates. In Study 1, the candidate applied for a Project Manager position. However, to ensure the appropriateness of the target role for our experimental design, we conducted a pilot study (*N* = 50) prior to Study 2. Participants in the pilot study evaluated a vignette describing a Project Coordinator or a Project Manager among other positions and rated the extent to which each role required competence, warmth (sociability), and status (high vs. low). Based on the results, the Project Coordinator role was selected for Studies 2 and 3, as it was perceived to require a balanced combination of competence and warmth, involved a moderate level of status, and was regarded as gender-neutral—that is, equally suitable for both male and female applicants.

Studies 1 and 2 examined how work setting (teleworking versus office) interacts with a) applicant's age (younger versus older) and b) applicant's gender (female versus male) to shape applicant's warmth and competence perceptions and subsequently hiring recommendations. Building on the findings of Studies 1 and 2, Study 3 was designed to further explore the interaction between work setting and applicant age specifically, omitting gender from the research model to allow for a more focused analysis. Accordingly, Study 3 did *not* test Hypotheses 2a, 2b and 4a, 4b that focused on the work-setting by gender interactions. Instead, for the sake of a more nuanced age manipulation, Study 3 included three applicant age groups (young, middle-aged, and older) in the age manipulations. Moreover, while Studies 1 and 2 largely relied on non-recruiter participants mostly, Study 3 primarily involved professional recruiters working in organizations, supplemented by a smaller number of non-recruiter participants.

**Open science and transparency practices.** Both Study 1 (https://osf.io/a9utq/overview; DOI number: 10.17605/OSF.IO/A9UTQ) and Study 2 (https://osf.io/wq52z/overview; DOI number: 10.17605/OSF.IO/WQ52Z) were preregistered in Open Science Framework. The preregistrations include hypotheses concerning additional variables not addressed in this manuscript (e.g., goals). To maintain clarity and focus, we chose to exclude these from the main text. However, all related data are openly available on the Open Science Framework: https://doi.org/10.17605/OSF.IO/FQ2PE. Moreover, we initially hypothesized that applicant gender would moderate the interaction between work setting and age in predicting hiring recommendations. However, as none of the studies revealed significant three-way interactions—and in light of the mixed findings in the literature regarding gender's role in shaping age-related effects (see, for instance, Fousiani, Scheibe, Griep, & El Khawli, 6)—we chose not to include this three-way interaction in the present manuscript or formulate specific hypotheses about it.

## Study 1

### Method

**Design and participants.** This study had a between-subjects 2 (work setting: teleworking versus office) x 2 (applicant age: younger versus older) x 2 (applicant gender: female versus male) design. We recruited 343 Dutch participants in total. Sensitivity power analysis showed that our sample yielded 80% power to detect a small to medium effect size (*f* = 0.15) at an alpha level of .05. Of the participants, 65.31% were female, and the mean age was 34.14 years (*SD* = 14.42). 80.76% had obtained a higher education degree (university degree or higher). We recruited participants from diverse occupational backgrounds, including those in full-time employment (39.94%), part-time employment (17.49%), full-time studies (35.86%), and part-time studies (2.33%). A small percentage (4.37%) were neither employed nor enrolled in education.

**Procedure and experimental manipulations.** Participants were recruited through the social networks of university students who assisted with data collection. Questionnaires were administered in Dutch (the participants' mother

language), and were completed online using Qualtrics. Participation lasted approximately 15 minutes. Data collection took place in May till July 2023. Prior to the data collection, written ethical approval was granted by the Scientific and Ethical Review Board of the Faculty of Behavioural and Movement Sciences at Free University of Amsterdam (reference number VCWE-2023-003R1). The board confirmed that the research complies with institutional ethical guidelines. All participants provided their active informed consent prior to participation. Information regarding the ethical, cultural, and scientific considerations specific to inclusivity in global research is included in the Supporting Information (S1 File).

We followed a procedure similar to Fousiani, Sypes, and Armenta [1] where participants read the description of a job vacancy seeking for a "project manager" to manage a team. Then, they were asked to take the perspective of an HR manager, who was supposed to hire one of the applicants that had applied. After being requested to immerse themselves into this role, participants were presented with relevant information regarding the work setting and job requirements for this position. Subsequently, participants were randomly assigned to one of the two work setting conditions (teleworking versus office). Following the work setting manipulations used by Fousiani, Sypes, and Armenta [1], participants in the office condition read that the selected employee would work full-time from an office, whereas those in the teleworking condition were informed that the employee would work full-time remotely (e.g., from home). More specifically, the office condition read: *"... the successful candidate will work 35-40 hours per week from the company office. Physical presence of the candidate in the office is mandatory. All kinds of formal or informal meetings, business discussions, and social interactions with team members and customers will be face-to-face. This means that the successful candidate will only meet other people face-to-face while no online interactions will take place. In other words, the successful candidate will be working with people that he or she will be able to meet in real life. Teleworking is not an option in the company."*

The teleworking condition read: *"… the successful candidate will work 35-40 hours per week remotely (e.g., from home). Physical presence of the candidate in the company is not necessary. All kinds of formal or informal meetings, business discussions, and social interactions with team members and customers will be completely online. This means that the successful candidate will only meet other people virtually while no face-to-face-interactions will take place. In other words, the successful candidate will be working with people that he or she might never be able to meet in real life. Working from a company office is not an option in the company."*

Subsequently, participants were informed that four applicants had been shortlisted for the position and that they would be randomly presented with a brief background description of one of these applicants. Then, they read a brief description of the applicant's background, in which the applicant's age and gender were manipulated. Age was manipulated by presenting the applicant as either 55 (older) or 30 years old (younger), an age threshold that is commonly used in the age literature [7,25]. Gender was manipulated using a typically Dutch, gender-salient name (Mr. Jan de Vries versus Ms. Johanna de Vries). To strengthen our manipulations, the background information was accompanied by a picture matching the applicant's age and gender. The chosen pictures were rated as average in terms of attractiveness and likability in a pre-test using a standardized photo set [38]. Any other background information about the applicant was kept constant. The specific texts read:

*"Mr. Jan de Vries [Ms. Johanna de Vries] was born in 1968 [1993] and he [she], therefore, is 55 years old [30 years old]. This is the candidate, Mr Jan de Vries [Ms. Johanna de Vries]* (the applicant's picture followed)*. Mr. Jan de Vries [Ms. Johanna de Vries] has considerable experience in Project Management. Based on his [her] CV, Mr de Vries [Ms. de Vries] fulfils the job requirements.* The complete vignettes and instructions can be found in S2 File (S2_Online supplemental material).

Participants then answered the manipulation checks about work setting and the applicant age and gender. After that, participants answered questions including perceived applicant's warmth and competence, and hiring recommendation. At the end of the study, the participants completed the demographic questions and were debriefed. No monetary compensation was offered to the participants.

**Measures. Manipulation Checks** To check whether our manipulation worked as intended, participants first answered two questions about work settings (i.e., "Based on what you just read, the successful candidate will work… 1) Completely remotely from home; 2) Always from the company office"; 1 = *Not at all true*, 7 = *Completely true*). For the age and gender manipulations, participants were asked to indicate the applicant's age and gender by selecting whether the applicant was (1) 30 or 55 years old, and (2) female or male.

**Perceived Warmth and Competence of the Applicant** We used a 14-item scale to measure perceived warmth (sociability) and competence of the applicant based on the Agency and Communion Scale of Abele and Wojciszke ( [39]; see also Fousiani, Sypes, and Armenta [1]) for the use of a similar scale). Seven items measured perceived warmth of the applicant (...Mr de Vries [Ms. de Vries], while at work, would be likable, supportive, helpful, social, friendly, kind, and warm; α = 0.93) and seven items measured perceived competence of the applicant (Mr de Vries [Ms. de Vries], while at work, would be active, skillful, capable, intelligent, competent, energetic, and efficient; 1 = *Not at all*, 7 = *To a great extent;* α = 0.89).

**Hiring Recommendation** We used the 2-item measure of Fousiani, Sypes, and Armenta [1] to assess the extent to which participants would recommend the applicant for the job: "Would you recommend that Mr de Vries [Ms. de Vries] be hired?" (1 = *Absolutely not*, 7 = *Absolutely yes*); "What is the likelihood that you would recommend Mr de Vries [Ms. de Vries] for hiring by your company?" (1 = *Very unlikely*, 7 = *Very likely;* α = 0.91).

**Control Variables** We controlled for participants' age and gender, given that we manipulated both age and gender of the applicants and considered prior research showing a significant effect of these evaluator variables in hiring decisions [9]. Participants reported their age in years and selected their gender from four options*: Female, Male, Other,* or *Prefer not to say.*

## Results

The means, standard deviations, and intercorrelations of the study variables are displayed in Table 1.

**Manipulation checks and preliminary analyses.** To test whether our manipulation checks were successful, we ran a multivariate regression with work setting (teleworking versus office) as the fixed variable and the manipulation check items for work setting, applicant age (1 = *30 Years old*, 2 = *55 Years old*), and applicants' gender (1 = *Male*, 2 = *Female*) as dependent variables. The results showed that the main effect of the teleworking setting on the corresponding perception was significant, $F (1, 341) = 806.95$, $p < .001$, $\eta_p^2 = 0.70$. Participants were more likely to perceive that the applicants

**Table 1. Descriptive statistics and correlations between the study variables (studies 1 and 2).**

|  | Study 1 | | Study 2 | | 1 | 2 | 3 | 4 | 5 | 6 | 7 | 8 |
|---|---|---|---|---|---|---|---|---|---|---|---|---|
|  | M | SD | M | SD |  |  |  |  |  |  |  |  |
| 1. Work setting | 1.50 | 0.50 | 1.51 | 0.50 |  | .01 | −.02 | −.04 | −.01 | −.05 | −.01 | −.06 |
| 2. Applicant age | 1.45 | 0.50 | 1.51 | 0.50 |  |  | −.03 | .17** | .07 | .06 | .03 | −.04 |
| 3. Applicant gender | 1.50 | 0.50 | 1.50 | 0.50 | .02 |  |  | −.11* | −.15** | −.12* | −.05 | −.02 |
| 4. Warmth | 4.24 | 0.98 | 4.36 | 1.06 | .18*** | .06 |  |  | .52*** | .45*** | −.08 | −.04 |
| 5. Competence | 4.47 | 0.95 | 4.60 | 1.00 | −.02 | −.08 | .61*** |  |  | .66*** | −.05 | .05 |
| 6. Hiring recommendation | 4.24 | 1.31 | 4.26 | 1.49 | −.16*** | −.01 | .51*** | .68*** |  |  | −.07 | .03 |
| 7. Participants' age | 34.14 | 14.42 | 35.64 | 15.40 | .04 | −.02 | −.19*** | −.10* | −.08 |  |  | .05 |
| 8. Participants' gender | 1.39 | 0.61 | 1.38 | 0.54 | .07 | .01 | .05 | −.04 | .01 | −.05 |  |  |

Notes. Work setting was coded as follows: 1 = *Office,* 2 = *Teleworking*; Applicant age was coded as follows: 1 = *Younger,* 2 = *Older*; Applicant gender was coded as follows: 1 = *Female,* 2 = *Male*; Correlations of Study 1 are above the diagonal and correlations of Study 2 below the diagonal; *p < .05, **p < .01, ***p < .001.

will work completely remotely from home in the teleworking setting condition ($M = 6.23$, $SD = 1.57$), as compared to the office setting condition ($M = 1.57$, $SD = 1.48$). Similarly, the office setting manipulation had a significant main effect on corresponding perception, $F (1,341) = 1034.78$, $p < .001$, $\eta_p^2 = 0.75$. Specifically, participants were more likely to perceive that the applicants will always work from the company office in the office setting condition ($M = 6.48$, $SD = 1.17$), as compared to the teleworking setting condition ($M = 1.67$, $SD = 1.57$).

For the manipulation checks of applicant age and gender, participants perceived both manipulations as intended, with 100% alignment between the assigned conditions and their responses. Taken together, these results indicated that participants perceived the manipulations as intended.

We then conducted a confirmatory factor analysis (CFA) with Mplus 7.00 [40], including the applicant's warmth, competence (the mediators), and hiring recommendation (the dependent variable) to test the fitness of our model. We specified correlated errors between largely similar items "energetic" and "active," "skillful" and "competent," and "competent" and "capable", an approach suggested by Bollen and Lennox (1991). The results indicated good model fit ($\chi2 = 328.60$, $df = 98$, $p < .001$; RMSEA = .08, [$CI_{90} = .07;.09$]; CFI = .94; SRMR = .08).

Considering the strong positive correlation between warmth and competence (see Table 1) we conducted separate analyses for each mediator in order to identify their unique contribution to the model. When warmth and competence were entered as simultaneous mediators, their effects on hiring recommendation were no longer significant. This is likely due to their strong correlation and their interdependent nature in social perception. Therefore, consistent with prior research (e.g., 11,39), we examined them separately to avoid redundancy and ensure conceptual clarity.

**Work setting, applicant age, and applicant gender on hiring recommendation through perceived warmth of the applicant.** To test whether a) work setting and age, and b) work setting and gender interact in predicting hiring recommendation through perceived warmth, we ran a moderated mediation model with PROCESS macro (Model 10; [41]). Work setting (1 = office, 2 = teleworking) was the independent variable, applicant age (1 = Younger, 2 = Older) and gender (1 = Male, 2 = Female) were the two moderators, applicant's warmth was the mediator, and hiring recommendation was the dependent variable. Participants' age and gender were included as the control variables. The overall model was significant, $R^2 = .22$, $F (8, 334) = 12.07$, $p < .001$.

In line with Hypothesis 1a, suggesting that work setting interacts with applicant age in predicting hiring recommendation, results showed that indeed, the interaction between work setting and applicant age on hiring recommendation was significant ($\Delta R^2 = .01$, $F (1,334) = 6.00$, $p = .015$). The simple slope analysis showed that recruiters were more likely to recommend older applicants for hire when the role involved working from an office, compared to teleworking ($B = -0.45$, $SE = .21$, $p = .034$; $CI_{95\%}$ [- 0.86; −0.03]) (see Fig 3). However, contrary to Hypothesis 3a, the simple effect for the hypothesized preference for younger applicants in teleworking, as compared to office, was not significant ($B = 0.12$, $SE = .19$, $p = .532$; $CI_{95\%}$ [−0.26; 0.50]). Moreover, while the applicant's warmth (mediator) was positively and significantly related to hiring recommendation ($B = 0.59$, $SE = .07$, $p < .001$; $CI_{95\%}$ [0.46; 0.72]), the interaction between work setting and applicant age on perceived warmth was not significant ($\Delta R^2 = .00$, $F (1,335) = 0.26$, $p = .608$) (see Table 2). Thus, Hypothesis 1b was not supported.

Unexpectedly, neither the interaction between work setting and applicant gender on hiring recommendation ($\Delta R^2 = .00$, $F (1,334) = 0.01$, $p = .908$), nor perceived applicant's warmth ($\Delta R^2 = .00$, $F (1,335) = 0.14$, $p = .710$), was significant (see Table 2). Therefore, Hypotheses 2a, 2b, and 4a were not supported.

**Work setting, applicant age, and applicant gender on hiring recommendation through perceived competence of the applicant.** To test the hypothesized moderated mediation model with perceived competence as the mediator, we again ran the PROCESS macro (Model 10) using the same specifications as above, with the mediator replaced by perceived competence. The overall model was significant, $R^2 = .44$, $F (8, 334) = 32.88$, $p < .001$. While the applicant's competence was positively and significantly related to hiring recommendation ($B = 0.89$, $SE = .06$, $p < .001$; $CI_{95\%}$ [0.78; 1.00]), neither the interaction between work setting and applicant age ($\Delta R^2 = .00$, $F (1,335) = 1.45$, $p = .229$), nor between

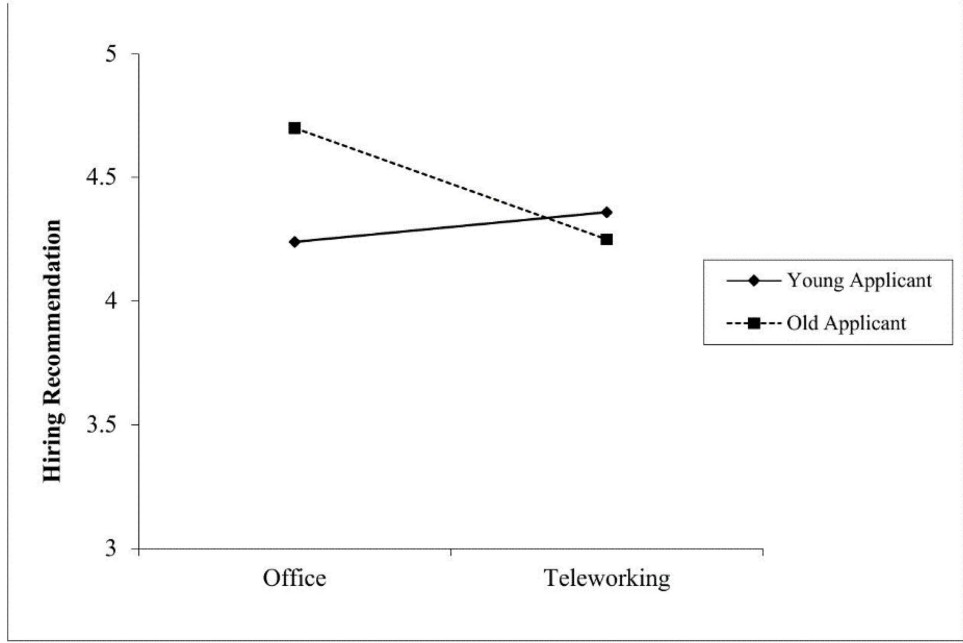

**Fig 3. Relationship Between Work Setting and Hiring Recommendation as Moderated by Applicant Age (Study 1).** Note. Hiring recommendation was assessed in a 7-point Likert scale (1 = *Very unlikely*, 7 = *Very likely*).

work setting and applicant gender ($\Delta R^2$ = .00, $F$ (1,335) = 0.00, $p$ = .976) on perceived competence was significant (see Table 3). Therefore, Hypotheses 3b and 4b were not supported.

## Discussion

Study 1 investigated the interaction between work setting (teleworking versus office-) and applicants' age (younger versus older) and gender (female versus male) in predicting hiring recommendations. The results provided support for Hypothesis 1a, showing that recruiters were more likely to recommend older applicants for hire when the position involved working from an office compared to teleworking. This finding aligns with the idea that older workers are perceived as more suitable for roles that require face-to-face collaboration and social interaction, which are more typical of office settings. However, contrary to Hypothesis 3a, the simple slope analysis for the hypothesized preference of younger applicants in telework settings was not significant. This suggests that, while older applicants are preferred in office-based than teleworking settings, the assumed preference for younger applicants in teleworking as compared to in-office settings may be less pronounced. Finally, contrary to Hypotheses 2a and 4a, work setting did not interact with applicant gender, showing that applicant gender does not significantly influence hiring recommendations across work settings. Unexpectedly, the work setting did not interact with either applicant age or gender in predicting perceived warmth or competence. As a result, Hypotheses 1b, 2b, 3b, and 4b were not supported, and the studies did not uncover the anticipated mediating mechanisms underlying the hypothesized effects of work setting on hiring recommendations.

Study 1 operationalized younger and older applicants as 30 and 55 years old, respectively, following the approach of [7]. However, the definition of "older age" varies across the literature, with some studies adopting a higher age threshold (e.g., Fousiani, Scheibe, & Michelakis, 6). To address this variability and test the robustness of our findings, we conducted a follow-up study (Study 2) in which we increased the age gap between conditions by defining older applicants as 60 years old instead of 55 and younger applicants as 28 years old. This allowed us to examine whether a wider age contrast

**Table 2. Relationship between work setting and hiring recommendation through perceived applicant's warmth as moderated by applicant age (Study 1).**

| Predictor | B | SE | p | 95% CI |
|---|---|---|---|---|
| Mediator: Perceived Applicant's Warmth | | | | |
| Work setting | −0.36 | 0.46 | 0.44 | −1.26; 0.54 |
| Older (vs. younger) applicant | 0.16 | 0.33 | 0.63 | −0.50; 0.82 |
| Work setting x older (vs. younger) applicant | 0.11 | 0.21 | 0.61 | −0.31; 0.52 |
| Male (vs. female) applicant | −0.34 | 0.33 | 0.31 | −0.99; 0.31 |
| Work setting x male (vs. female) applicant | 0.08 | 0.21 | 0.71 | −0.33; 0.49 |
| Participants' age | −0.01 | 0.00 | 0.09 | −0.01; 0.00 |
| Participants' gender | −0.06 | 0.09 | 0.51 | −0.23; 0.11 |
| Dependent Variable: Hiring Recommendation | | | | |
| Perceived applicant's warmth | 0.59 | 0.07 | 0.00 | 0.46; 0.72 |
| Work setting | 0.86 | 0.55 | 0.12 | −0.23; 1.95 |
| Older (vs. younger) applicant | 0.92 | 0.41 | 0.02 | 0.12; 1.71 |
| Work setting x older (vs. younger) applicant | −0.63 | 0.26 | 0.01 | −1.13; −0.12 |
| Male (vs. female) applicant | −0.15 | 0.40 | 0.71 | −0.94; 0.64 |
| Work setting x male (vs. female) applicant | −0.03 | 0.25 | 0.91 | −0.53; 0.47 |
| Participants' age | 0.00 | 0.00 | 0.30 | −0.01; 0.00 |
| Participants' gender | 0.13 | 0.11 | 0.22 | −0.08; 0.34 |
| Conditional Indirect Effects of Work Setting at Levels of Applicant Gender and Age | | | | |
| Applicant Gender | Applicant Age | B | SE | 95% CI |
| Female | Young | −0.10 | 0.11 | −0.34; 0.10 |
| Females | Old | −0.06 | 0.11 | −0.26; 0.16 |
| Male | Young | −0.04 | 0.11 | −0.25; 0.18 |
| Male | Old | 0.01 | 0.11 | −0.19; 0.22 |
| Conditional Direct Effects of Work Setting at Levels of Applicant Gender and Age | | | | |
| Applicant Gender | Applicant Age | B | SE | 95% CI |
| Female | Young | 0.20 | 0.22 | −0.22; 0.63 |
| Female | Old | 0.18 | 0.21 | −0.24; 0.59 |
| Male | Young | −0.42 | 0.22 | −0.86; 0.02 |
| Male | Old | −0.45 | 0.23 | −0.90; 0.00 |

Note. Work setting was coded as follows: 1 = *Office*, 2 = *Teleworking*; Applicant age was coded as follows: 1 = *Younger,* 2 = *Older*; Applicant gender was coded as follows: 1 = *Female*, 2 = *Male*; CI = confidence interval.

would yield similar (or even stronger) patterns of results. In this follow-up study, work setting and gender were manipulated similarly to Study 1.

## Study 2

### Method

**Design and participants.** This study had a between-subjects 2 (work setting: teleworking versus office) x 2(applicant age: younger versus older) x 2 (applicant gender: female versus male) design. Questionnaires were administered in Dutch and were completed online using Qualtrics. We recruited 449 participants, the majority of whom were Dutch (90.42%) while the rest were Dutch-speaking internationals living in the Netherlands. Sensitivity power analysis showed that our sample yielded 80% power to detect a small to medium effect size ($f = 0.13$) at an alpha level of.05. 62.81% of

**Table 3. Relationship between work setting and hiring recommendation through Perceived applicant's competence as moderated by applicant age (Study 1).**

| Predictor | B | SE | p | 95% CI |
|---|---|---|---|---|
| **Mediator: Perceived Applicant's Competence** | | | | |
| Work setting | 0.33 | 0.45 | 0.46 | −0.55; 1.20 |
| Older (vs. younger) applicant | 0.50 | 0.33 | 0.13 | −0.14; 1.14 |
| Work setting x older (vs. younger) applicant | −0.25 | 0.21 | 0.23 | −0.65; 0.16 |
| Male (vs. female) applicant | −0.30 | 0.32 | 0.36 | −0.93; 0.34 |
| Work setting x male (vs. female) applicant | 0.01 | 0.20 | 0.98 | −0.39; 0.41 |
| Participants' age | 0.00 | 0.00 | 0.22 | −0.01; 0.00 |
| Participants' gender | 0.08 | 0.08 | 0.32 | −0.08; 0.25 |
| **Dependent Variable: Hiring Recommendation** | | | | |
| Perceived applicant's competence | 0.89 | 0.06 | 0.00 | 0.78; 1.00 |
| Work setting | 0.36 | 0.47 | 0.45 | −0.57; 1.28 |
| Older (vs. younger) applicant | 0.56 | 0.35 | 0.10 | −0.12; 1.24 |
| Work setting x older (vs. younger) applicant | −0.34 | 0.22 | 0.12 | −0.77; 0.09 |
| Male (vs. female) applicant | −0.09 | 0.34 | 0.80 | −0.75; 0.58 |
| Work setting x male (vs. female) applicant | 0.01 | 0.21 | 0.96 | −0.41; 0.43 |
| Participants' age | 0.00 | 0.00 | 0.24 | −0.01; 0.00 |
| Participants' gender | 0.02 | 0.09 | 0.83 | −0.16; 0.19 |
| **Conditional Indirect Effects of Work Setting at Levels of Applicant Gender and Age** | | | | |
| Applicant Gender | Applicant Age | B | SE | 95% CI |
| Female | Young | 0.08 | 0.16 | −0.24; 0.37 |
| Female | Old | 0.08 | 0.14 | −0.19; 0.36 |
| Male | Young | −0.15 | 0.17 | −0.48; 0.19 |
| Male | Old | −0.14 | 0.16 | −0.46; 0.18 |
| **Conditional Direct Effects of Work Setting at Levels of Applicant Gender and Age** | | | | |
| Applicant Gender | Applicant Age | B | SE | 95% CI |
| Female | Young | 0.03 | 0.18 | −0.33; 0.39 |
| Female | Old | 0.04 | 0.18 | −0.31; 0.39 |
| Male | Young | −0.31 | 0.19 | −0.69; 0.06 |
| Male | Old | −0.30 | 0.19 | −0.68; 0.08 |

Note. Work setting was coded as follows: 1 = *Office,* 2 = *Teleworking*; Applicant age was coded as follows: 1 = *Younger,* 2 = *Older*; Applicant gender was coded as follows: 1 = *Female,* 2 = *Male*; CI = confidence interval.

the employees were female, and the mean age was 35.64 years (*SD* = 15.40). Of the participants, 86.41% had obtained a higher education degree (university degree or higher). Similar to Study 1, the sample included individuals from various occupational backgrounds: participants in permanent employment (44.10%), part-time employment (7.13%), self-employment (8.02%), unemployed (2.90%), students (30.29%), and retirees (6.24%). Six participants (1.34%) did not report their occupational status.

**Procedure, manipulation, and measures.** The procedure, manipulations, and measures in Study 2 were largely similar to those used in Study 1. However, as explained in the overview of the study section, this time, the younger applicant's age was set to 28 years old and the older applicant's age was set to 60 years olde (see also Fousiani, Scheibe, & Michelakis, 6; for the age increase in the older applicant condition). Accordingly, the applicants' pictures were changed to visually illustrate the age difference. Moreover, the target position was changed from Project Manager to Project Coordinator, a role considered to be more gender-neutral, in contrast to Study 1 (see overview of the current

research paragraph for a more detailed explanation of the role). The research material can be found in S2 File (S2_Online supplemental material). After being exposed to the manipulations, participants again answered questions similar to those used in Study 1, including perceived applicant's warmth (α = 0.93), competence (α = 0.90), hiring recommendation (α = 0.93), manipulation checks, and their demographics. Participants provided their written informed consent prior to completing the questionnaires. Data collection took place in February till May 2024. Similar to Study 1, prior to the data collection, written ethical approval was granted by the Scientific and Ethical Review Board of Free University of Amsterdam.

## Results

The means, standard deviations, and intercorrelations of the study variables are displayed in Table 1.

**Manipulation checks and preliminary analyses.** To test whether our manipulation checks were successful, we ran a multivariate regression with work setting (teleworking versus office) as the fixed variables and the manipulation check items for work setting (1 = *28 Years old*, 2 = *60 Years old*) as dependent variables. The results showed that the main effect of the teleworking setting on the corresponding perception was significant, $F(1,447) = 709.55$, $p < .001$, $\eta_p^2 = 0.61$. Participants were more likely to perceive that the applicants will work completely remotely from home in the teleworking setting condition ($M = 6.18$, $SD = 1.61$), as compared to the office setting condition ($M = 1.93$, $SD = 1.76$). Similarly, the office setting manipulation had a significant main effect on corresponding perception, $F(1,447) = 501.35$, $p < .001$, $\eta_p^2 = 0.53$. Specifically, participants were more likely to perceive that the applicants will always work from the company office in the office setting condition ($M = 5.95$, $SD = 1.75$), as compared to the teleworking setting condition ($M = 2.06$, $SD = 1.93$).

For the manipulation checks of applicant age and gender, participants perceived both manipulations as intended, with 100% alignment between the assigned conditions and their responses. Taken together, these results indicated that participants perceived the manipulations as intended.

We then conducted a confirmatory factor analysis (CFA) using Mplus 7.00 (40), which included applicants' warmth, competence (the mediators), and hiring recommendation (the dependent variable) to test the fitness of our model. Consistent with Study 1, we specified correlated errors between the same pairs of similarly worded items (cf., [42]). The results indicated acceptable model fit ($\chi 2 = 416.28$, $df = 98$, $p < 0.001$; RMSEA = .09, [$CI_{90} = .08; .09$]; CFI = .94; SRMR = .08).

Similar to Study 1, considering the strong positive correlation between warmth and competence (see Table 1) we conducted separate analyses for each mediator, in order to identify their unique contribution to the model.

**Work setting, applicant age, and applicant gender on hiring recommendation through perceived warmth of the applicant.** Similar to Study 1, to test whether work setting interacts with applicant age or applicant gender in predicting hiring recommendation, we ran a moderated mediation analysis using PROCESS macro (Model 10; (41)). Work setting (1 = *Office*, 2 = *Teleworking*) was the independent variable, applicant age (1 = *Younger*, 2 = *Older*) and gender (1 = *Male*, 2 = *Female*) were the moderators, applicant's warmth was the mediator, and hiring recommendation was the dependent variable. Similar to Study 1, participants' age and gender were included as the control variables. The overall model was significant, $R^2 = .34$, $F(8, 436) = 27.74$, $p < .001$.

In line with Hypothesis 1a, suggesting that work setting interacts with applicant age in predicting hiring recommendations, results showed that indeed, the interaction between work setting and applicant age on hiring recommendation was significant ($\Delta R^2 = .01$, $F(1,436) = 4.78$, $p = .029$). The simple slope analysis showed that recruiters were more likely to recommend older applicants for hire when the role involved working from an office, compared to teleworking ($B = -0.49$, $SE = .19$, $p = .012$; $CI_{95\%}$ [- 0.87; −0.11]) (see Fig 4). However, contrary to Hypothesis 3a, the simple effect for the hypothesized preference for younger applicants in teleworking, as compared to office, was not significant ($B = 0.11$, $SE = .20$, $p = .569$; $CI_{95\%}$ [- 0.28; 0.50]). Moreover, while the applicant's warmth (mediator) was positively and significantly related to hiring recommendation ($B = 0.80$, $SE = .06$, $p < .001$; $CI_{95\%}$ [0.68; 0.91]), the interaction between work setting and applicant

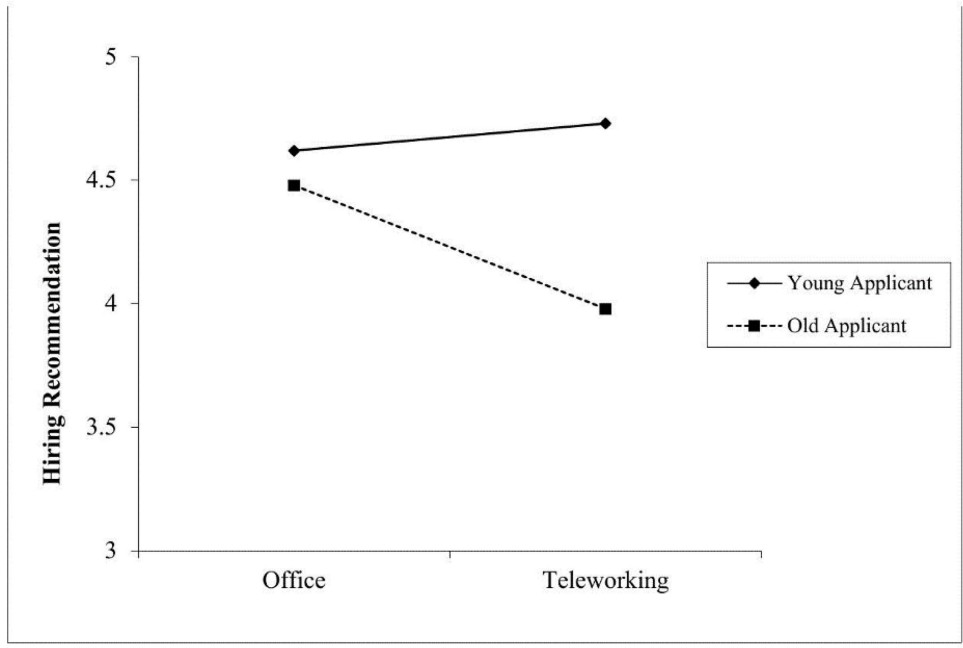

**Fig 4. Relationship Between Work Setting and Hiring Recommendation as Moderated by Applicant Age (Study 2).** Note. Hiring recommendation was assessed in a 7-point Likert scale (1 = *Very unlikely*, 7 = *Very likely*).

age on perceived warmth was not significant ($\Delta R^2$ = .00, $F$ (1,437) = 0.42, $p$ = .519) (see Table 4). Thus, Hypothesis 1b was not supported.

Unexpectedly, neither the interaction between work setting and applicant gender on hiring recommendation ($\Delta R^2$ = .00, $F$ (1,436) = 0.06, $p$ = .806), nor perceived applicant's warmth ($\Delta R^2$ = .00, $F$ (1,437) = 0.79, $p$ = .374), was significant (see Table 4). Therefore, Hypotheses 2a, 2b, and 4a were not supported.

**Work setting, applicant age, and applicant gender on hiring recommendation through perceived competence of the applicant.** To test the hypothesized moderated mediation model with perceived competence as the mediator, we again ran the PROCESS macro (Model 10) using the same specifications as above, with the mediator replaced by perceived competence. The overall model was significant, $R^2$ = .51, $F$ (8, 436) = 56.31, $p$ < .001. Similar to Study 1, while the applicant's competence was positively and significantly related to hiring recommendation ($B$ = 1.02, $SE$ = .05, $p$ < .001; $CI_{95\%}$ [0.92; 1.12]), neither the interaction between work setting and applicant age ($\Delta R^2$ = .00, $F$ (1,437) = 0.14, $p$ = .709), nor between applicant gender ($\Delta R^2$ = .00, $F$ (1,437) = 0.16, $p$ = .692) on perceived competence was significant (see Table 5). Therefore, Hypotheses 3b and 4b were not supported.

## Discussion

Study 2 aimed to replicate the findings of Study 1 while increasing the age gap between applicants in the experimental manipulation to align with research suggesting a higher age threshold (e.g., Fousiani, Scheibe, & Michelakis, 6). The results largely replicated those of Study 1: Work setting interacted with applicant age, such that older applicants received higher hiring recommendations in the office condition compared to the teleworking condition. Once again, applicant gender did not interact with work setting, suggesting that gender was not a significant factor in shaping hiring decisions across different work settings. Moreover, as in Study 1, work setting did not interact with applicant age or gender in predicting perceived warmth or competence. As a result, Study 2 also did not provide evidence supporting the proposed

**Table 4. Relationship Between Work Setting and Hiring Recommendation Through Perceived Applicant's Warmth as Moderated by Applicant Age (Study 2).**

| Predictor | B | SE | p | 95% CI |
|---|---|---|---|---|
| Mediator: Perceived Applicant's Warmth | | | | |
| Work setting | 0.36 | 0.42 | 0.40 | −0.47; 1.18 |
| Older (vs. younger) applicant | 0.61 | 0.31 | 0.05 | 0.00; 1.21 |
| Work setting x older (vs. younger) applicant | −0.13 | 0.19 | 0.52 | −0.51; 0.26 |
| Male (vs. female) applicant | 0.34 | 0.31 | 0.27 | −0.27; 0.94 |
| Work setting x male (vs. female) applicant | −0.17 | 0.19 | 0.37 | −0.55; 0.21 |
| Participants' age | −0.01 | 0.00 | 0.00 | −0.02; −0.01 |
| Participants' gender | 0.05 | 0.09 | 0.55 | −0.12; 0.23 |
| Dependent Variable: Hiring Recommendation | | | | |
| Perceived applicant's warmth | 0.80 | 0.06 | 0.00 | 0.68; 0.91 |
| Work setting | 0.56 | 0.50 | 0.27 | −0.43; 1.55 |
| Older (vs. younger) applicant | −0.01 | 0.37 | 0.98 | −0.74; 0.72 |
| Work setting x older (vs. younger) applicant | −0.51 | 0.23 | 0.03 | −0.96; −0.05 |
| Male (vs. female) applicant | −0.20 | 0.37 | 0.59 | −0.92; 0.53 |
| Work setting x male (vs. female) applicant | 0.06 | 0.23 | 0.81 | −0.40; 0.51 |
| Participants' age | 0.00 | 0.00 | 0.38 | 0.00; 0.01 |
| Participants' gender | 0.00 | 0.11 | 0.98 | −0.21; 0.22 |

Conditional Indirect Effects of Work Setting at Levels of Applicant Gender and Age

| Applicant Gender | Applicant Age | B | SE | 95% CI |
|---|---|---|---|---|
| Female | Young | 0.05 | 0.14 | −0.24; 0.32 |
| Female | Old | −0.09 | 0.14 | −0.37; 0.18 |
| Male | Young | −0.05 | 0.12 | −0.30; 0.19 |
| Male | Old | −0.19 | 0.13 | −0.44; 0.06 |

Conditional Direct Effects of Work Setting at Levels of Applicant Gender and Age

| Applicant Gender | Applicant Age | B | SE | 95% CI |
|---|---|---|---|---|
| Female | Young | 0.11 | 0.20 | −0.29; 0.50 |
| Female | Old | 0.16 | 0.20 | −0.24; 0.56 |
| Male | Young | −0.40 | 0.20 | −0.80; 0.00 |
| Male | Old | −0.34 | 0.20 | −0.73; 0.05 |

Note. Work setting was coded as follows: 1 = *Office*, 2 = *Teleworking*; Applicant age was coded as follows: 1 = *Younger,* 2 = *Older*; Applicant gender was coded as follows: 1 = *Female,* 2 = *Male*; CI = confidence interval.

mediating role of warmth and competence in explaining the effects of work setting and applicant demographics on hiring recommendations.

Given the consistent lack of gender effects in both studies, Study 3 focused more closely on applicant age. It extended the design by introducing a three-level age manipulation (younger, middle-aged, and older applicants), allowing for a more nuanced investigation of age-related effects. This refinement was based on the possibility that applicant gender may have introduced variability that masked the impact of work setting on perceptions of warmth or competence across age groups. Indeed, the literature on the role of gender in how employed individuals are perceived is inconsistent, with some research showing that gender stereotypes can amplify age-related biases (e.g., older women being penalized more than older men), while other studies find minimal or no gender effects in organizational contexts ( [43]; for the inconsistent role of gender, see also 6). By removing gender as a variable in Study 3, we aimed to reduce this potential source of noise and

**Table 5.  Relationship Between Work Setting and Hiring Recommendation Through Perceived Applicant's Competence as Moderated by Applicant Age (Study 2).**

| Predictor | B | SE | p | 95% CI |
|---|---|---|---|---|
| Mediator: Perceived Applicant's Competence | | | | |
| Work setting | −0.04 | 0.41 | 0.92 | −0.85; 0.77 |
| Older (vs. younger) applicant | 0.11 | 0.30 | 0.72 | −0.49; 0.70 |
| Work setting x older (vs. younger) applicant | −0.07 | 0.19 | 0.71 | −0.44; 0.30 |
| Male (vs. female) applicant | −0.29 | 0.30 | 0.34 | −0.88; 0.30 |
| Work setting x male (vs. female) applicant | 0.08 | 0.19 | 0.69 | −0.30; 0.45 |
| Participants' age | −0.01 | 0.00 | 0.03 | −0.01; 0.00 |
| Participants' gender | −0.09 | 0.09 | 0.32 | −0.26; 0.09 |
| Dependent Variable: Hiring Recommendation | | | | |
| Perceived applicant's competence | 1.02 | 0.05 | 0.00 | 0.92; 1.12 |
| Work setting | 0.88 | 0.43 | 0.04 | 0.03; 1.74 |
| Older (vs. younger) applicant | 0.37 | 0.32 | 0.25 | −0.26; 0.99 |
| Work setting x older (vs. younger) applicant | −0.53 | 0.20 | 0.01 | −0.93; −0.14 |
| Male (vs. female) applicant | 0.37 | 0.32 | 0.25 | −0.26; 0.99 |
| Work setting x male (vs. female) applicant | −0.16 | 0.20 | 0.43 | −0.55; 0.24 |
| Participants' age | 0.00 | 0.00 | 0.88 | −0.01; 0.01 |
| Participants' gender | 0.14 | 0.09 | 0.15 | −0.05; 0.32 |
| Conditional Indirect Effects of Work Setting at Levels of Applicant Gender and Age | | | | |
| Applicant Gender | Applicant Age | B | SE | 95% CI |
| Female | Young | −0.04 | 0.18 | −0.39; 0.31 |
| Female | Old | 0.04 | 0.16 | −0.27; 0.35 |
| Male | Young | −0.11 | 0.16 | −0.43; 0.21 |
| Male | Old | −0.03 | 0.17 | −0.35; 0.29 |
| Conditional Direct Effects of Work Setting at Levels of Applicant Gender and Age | | | | |
| Applicant Gender | Applicant Age | B | SE | 95% CI |
| Female | Young | 0.19 | 0.17 | −0.15; 0.53 |
| Female | Old | 0.03 | 0.18 | −0.31; 0.38 |
| Male | Young | −0.34 | 0.17 | −0.69; 0.00 |
| Male | Old | −0.50 | 0.17 | −0.84; −0.17 |

Note. Work setting was coded as follows: 1 = *Office*, 2 = *Teleworking*; Applicant age was coded as follows: 1 = *Younger,* 2 = *Older*; Applicant gender was coded as follows: 1 = *Female,* 2 = *Male*; *CI* = confidence interval.

more precisely examine how applicant age shapes warmth and competence perceptions and subsequently hiring recommendations across different work settings.

## Study 3

### Method

**Design and participants.**  This study had a 2 (work setting: teleworking versus office) x 3(applicant age: younger versus middle-aged versus older) between-subjects experimental design. Participants were recruited in two ways: The majority of participants (professional recruiters) were recruited via Prolific ($n$ = 203; 57.18%), while some participants (non-recruiters) were recruited through the university lab ($n$ = 152; 42.82%), resulting in a total sample that included both individuals from the working population (professional recruiters) and students. Questionnaires were administered

in English and were completed online using Qualtrics. We recruited a total of 355 participants and excluded those who 1) indicated that they had indicated they provided invalid data ($n = 7$); 2) failed our attention checks ($n = 8$); 3) completed the survey in an unreasonably short amount of time (less than three minutes; $n = 3$); 4) failed the manipulation check by selecting an applicant age inconsistent with their assigned condition ($n = 9$). The final sample consists of 328 participants. Sensitivity power analysis showed that our sample yielded 80% power to detect a small to medium effect size ($f = 0.17$) at an alpha level of.05. Of the participants, 54.90% were female, and the mean age was 31.64 years ($SD = 13.68$). 59.05% had obtained a higher education degree (university degree or higher). Sixty-four (18.99%) of the participants were British, one hundred and eighteen (35.01%) were American, seventy-eight (23.15%) were Dutch, with the rest from other countries. Data collection took place in March till May, 2025. Similar to Studies 1 and 2, prior to the data collection, written ethical approval was granted by the Scientific and Ethical Review Board of Free University of Amsterdam.

**Procedure and manipulation.** Participants were first randomly assigned to one of three experimental conditions, manipulating work settings (teleworking versus office), similar to those used in Studies 1 and 2. Next, they were informed that they would review a description of one shortlisted applicant. The procedure was largely similar to that of Studies 1 and 2. However, unlike Studies 1 and 2, gender was not manipulated in this study; therefore, we did not provide any picture of the applicant and referred to the individual as "Candidate A" to minimize potential gender bias associated with names. Participants were then randomly assigned to one of three age conditions (younger: 28 years old, middle-aged: 45 years old, and older: 60 years old applicants).

After the manipulation check, participants answered questions including perceived applicant's warmth and competence, hiring recommendation, home-work interference, and perceived digital skills of the applicant. As in Studies 1 and 2, all participants provided their written informed consent prior to completing the questionnaires. At the end of the study, the participants completed the demographic questions, were debriefed, and paid (7 BP per hour).

**Measures.** Manipulation checks for work setting and applicant age followed those in Studies 1 and 2, with the addition of a 45-year-old option for the age manipulation check. The measures for perceived applicant's warmth ($α = 0.92$), competence ($α = 0.87$), and hiring recommendation ($α = 0.92$) were consistent with those used in Studies 1 and 2, with minor adaptations made to the instructions to suit the specific context of Study 3.

**Control Variables** Similar to Studies 1 and 2 we controlled for participants' age and gender. Given that multiple nationalities took part in this study, we also controlled for participant nationality. Moreover, in this study, we considered the role of home-work interference and perceived digital competency could cause potential differential treatment of applicants in recruitment decisions [23,26,27]. Therefore, we also controlled for these two variables.

***Home-work Interference*** A six-item scale [44] was used to measure to what extent participants perceived the job requirement and applicant's home life would interfere with each other (e.g., "The job requirements of this position would interfere with Candidate A's responsibilities at home, such as cooking, shopping, child care, yard work, and house repairs";1 = *Not at all*, 7 = *To a great extent*; $α = 0.93$).

***Perceived Digital Competency*** A three-item scale was used to assess participants' perceptions of the applicant's digital skills (e.g., "Candidate A is likely to make good use of digital technologies for the communication with his/her colleagues"; [45]; 1 = *Strongly disagree*, 7 = *Strongly agree*; $α = 0.82$).

## Results

The means, standard deviations, and intercorrelations of the study variables are displayed in Table 6.

**Manipulation checks and preliminary analyses.** To test whether our manipulation checks were successful, we ran a multivariate regression with work setting (teleworking versus office) as the fixed variable, and the manipulation checks for work setting as dependent variables. The results showed that the main effect of the teleworking setting on the corresponding perception was significant, $F (1, 326) = 1094.93$, $p < .001$, $\eta_p^2 = 0.77$. Participants were more likely to perceive that the applicants will work completely remotely from home in the teleworking setting condition ($M = 6.47$,

**Table 6. Descriptive Statistics and Correlations between the Study Variables (Study 3).**

| | M | SD | 1 | 2 | 3 | 4 | 5 | 6 | 7 | 8 | 9 | 10 |
|---|---|---|---|---|---|---|---|---|---|---|---|---|
| 1. Work setting | 1.49 | 0.50 | | 0.02 | 0.04 | 0.04 | −0.05 | −0.03 | 0.03 | 0.03 | .14*** | −.16** |
| 2. Applicant age | 2.01 | 0.82 | | | −0.06 | −.21*** | −.30*** | 0.00 | 0.04 | 0.05 | −.50*** | 0.06 |
| 3. Warmth | 5.19 | 0.90 | | | | .67*** | .41*** | .17** | .31*** | 0.03 | .33*** | −.15** |
| 4. Competence | 5.40 | 0.86 | | | | | .68** | .13* | .20*** | 0.00 | .45*** | −.26*** |
| 5. Hiring recommendation | 5.32 | 1.16 | | | | | | .11* | 0.09 | −0.07 | .49*** | −.31*** |
| 6. Participants' age | 31.40 | 13.65 | | | | | | | .23*** | 0.02 | .20*** | −0.03 |
| 7. Participants' nationality | 2.50 | 1.15 | | | | | | | | −0.07 | .20*** | 0.10 |
| 8. Participants' gender | 1.56 | 0.50 | | | | | | | | | −0.07 | −0.06 |
| 9. Perceived digital skills of the applicant | 4.67 | 1.42 | | | | | | | | | | −.24*** |
| 10. Home-office interference | 3.35 | 1.41 | | | | | | | | | | |

Notes. Work setting was coded as follows: 1 = *Office*, 2 = *Teleworking*; Applicant age was coded as follows: 1 = *Younger*, 2 = *Middle-aged*, 3 = *Older*;
*p < .05, **p < .01, ***p < .001.

$SD = 1.35$), as compared to the office setting condition ($M = 1.49$, $SD = 1.38$). Similarly, the office setting manipulation had a significant main effect on corresponding perception, $F(1, 326) = 861.59$, $p < .001$, $\eta_p^2 = 0.73$. Specifically, participants were more likely to perceive that the applicants will always work from the company office in the office setting condition ($M = 6.48$, $SD = 1.33$), as compared to the teleworking setting condition ($M = 1.68$, $SD = 1.63$).

For the manipulation checks of applicant age, we excluded participants who selected an applicant age inconsistent with their assigned condition ($n = 9$). As a result, in the final sample there was 100% alignment between participants' assigned conditions and their responses regarding applicant age.

We again conducted a confirmatory factor analysis (CFA) using Mplus 7.00 (40) to test the fitness of our model, which included applicants' warmth, competence (the mediators), hiring recommendation (the dependent variable), home-work interference, and perceived digital skills of the applicant (the controls). Consistent with Studies 1 and 2, we specified correlated errors between the same pairs of similarly worded items in the competence scale (Bollen & Lennox, 1991). The results indicated an acceptable model fit ($\chi^2 = 916.32$, $df = 262$, $p < 0.001$; RMSEA = 0.09, [$CI_{90} = 0.08$; 0.09]; CFI = 0.90; SRMR = 0.09).

Similar to Studies 1 and 2, considering the strong positive correlation between warmth and competence (see Table 1) we conducted separate analyses for each mediator, in order to identify their unique contribution to the model.

**Work setting, applicant age, and applicant gender on hiring recommendation through perceived warmth of the applicant.** To test whether work setting and age interact in predicting hiring recommendation through perceived warmth, we ran a moderated mediation model using the PROCESS macro (Model 8; (41). Work setting (1 = *Office*, 2 = *Teleworking*) was the independent variable, applicant's warmth was the mediator, and hiring recommendation was the dependent variable. Applicant's age was entered as the categorical moderator with three levels (1 = *Younger*, 2 = *Middle-aged*, 3 = *Older*), with the younger applicant group used as the reference category, consistent with Studies 1 and 2. Participants' age, gender, nationality, perceived home-work interference, and perceived digital skills of the applicant were included as the control variables. The overall model was significant, $R^2 = .41$, $F(11, 315) = 20.07$, $p < .001$.

Supporting Hypothesis 1a, the conditional direct effects of work setting on hiring recommendation were negative and only significant for older applicants ($B = −0.37$, $SE = .18$, $p = .039$; $CI_{95\%}$ [−0.71; −0.02]), which means that recruiters were more likely to recommend older applicants for hire when the role involved working from an office, compared to teleworking (see Fig 5). Consistent with previous studies and contrary to Hypothesis 3a, the conditional direct effects of work setting on hiring recommendation were not significant for younger applicants ($B = −0.31$, $SE = .18$, $p = .081$; $CI_{95\%}$ [−0.66; 0.04]) or mid-aged applicants ($B = −0.19$, $SE = .17$, $p = .278$; $CI_{95\%}$ [−0.53; 0.15]).

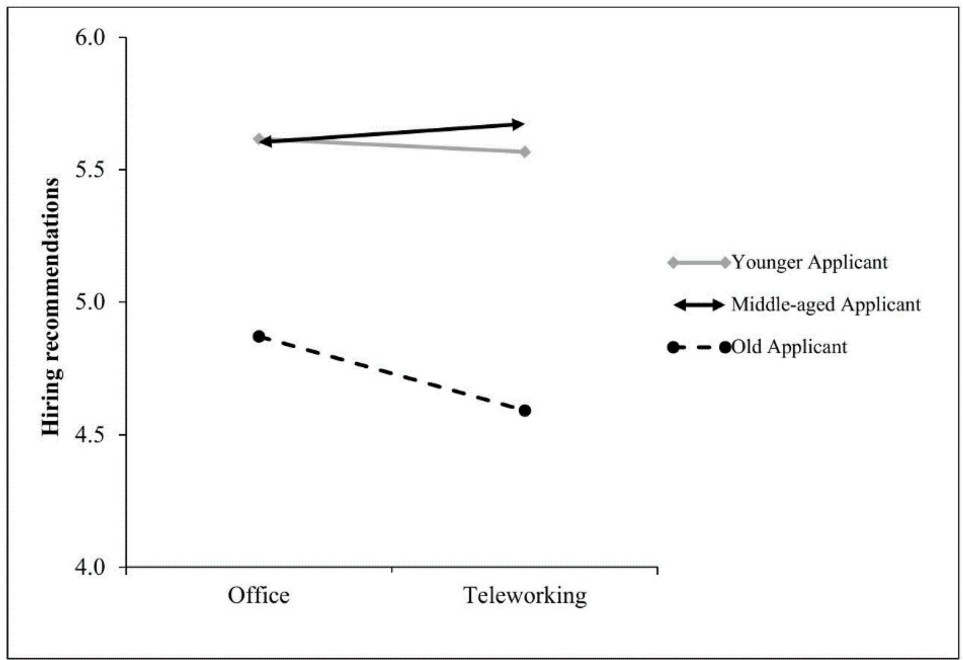

**Fig 5. Relationship Between Work Setting and Hiring Recommendation as Moderated by Applicant Age (Study 3).** Note. Hiring recommendation was assessed in a 7-point Likert scale (1 = *Very unlikely*, 7 = *Very likely*).

As shown in Table 7, the interaction between work setting and applicant age on perceived warmth was significant when comparing older versus younger applicants, but not when comparing middle-aged versus younger applicants. Simple effects analysis showed that only older applicants were perceived as warmer in office settings compared to telework settings, whereas this effect was not observed for middle-aged or younger applicants (see Table 7). Additionally, perceived applicant's warmth was positively and significantly related to hiring recommendation. Moreover, the indirect effect of work setting on hiring recommendation through perceived warmth was moderated by applicant age, when comparing younger versus older applicants (moderated mediation index = −0.17, $SE$ = 0.09; $CI_{95\%}$ [−0.38; −0.00]) (see Fig 6), but not when comparing younger versus middle-aged applicants (moderated mediation index = −0.03, $SE$ = 0.09; $CI_{95\%}$ [−0.20; 0.14]). Specifically, the indirect work setting – perceived applicant's warmth – hiring recommendation linkage was negative and only significant for older applicants (see Table 7). Our results hence provided support for Hypothesis 1b.

**Work setting, applicant age, and applicant gender on hiring recommendation through perceived competence of the applicant.** To test the hypothesized moderated mediation model with perceived competence as the mediator, we again ran the PROCESS macro (Model 8) using the same specifications as above, with the mediator replaced by perceived competence. The overall model was significant, $R^2$ = .56, $F(11, 315)$ = 36.31, $p < .001$. While the applicant's competence was positively and significantly related to hiring recommendation ($B$ = 0.73, $SE$ = 0.06, $p < .001$; $CI_{95\%}$ [0.62; 0.85]), the interactions between work setting and applicant age on perceived competence were not significant. The indirect effect of work setting on hiring recommendation through perceived competence was not moderated by applicant age (*younger versus older applicants:* moderated mediation index = −0.04, $SE$ = 0.16; $CI_{95\%}$ [−0.36; 0.27]; *younger vs middle-aged applicants:* moderated mediation index = −0.09, $SE$ = 0.15; $CI_{95\%}$ [−0.38; 0.21]), failing to provide support for Hypothesis 3b.

**Table 7. Relationship Between Work Setting and Hiring Recommendation through Perceived Applicant's Warmth as Moderated by Applicant Age (Study 3).**

| Predictor | B | SE | p | 95% CI |
|---|---|---|---|---|
| Mediator: Perceived Applicant's Warmth | | | | |
| Work setting | 0.13 | 0.16 | 0.42 | −0.19; 0.44 |
| Middle-aged (vs. younger) applicant | 0.12 | 0.35 | 0.73 | −0.56; 0.80 |
| Older (vs. younger) applicant | 0.86 | 0.36 | 0.02 | 0.16; 1.57 |
| Work setting x middle-aged (vs. younger) applicant | −0.08 | 0.22 | 0.72 | −0.52; 0.36 |
| Work setting x older (vs. younger) applicant | −0.45 | 0.22 | 0.05 | −0.89; −0.01 |
| Participants' age | 0.00 | 0.00 | 0.38 | 0.00; 0.01 |
| Participants' nationality | 0.20 | 0.04 | 0.00 | 0.12; 0.28 |
| Participants' gender | 0.08 | 0.09 | 0.37 | −0.10; 0.26 |
| Home-office interference | −0.06 | 0.03 | 0.06 | −0.13; 0.00 |
| Perceived digital skills of the applicant | 0.19 | 0.04 | 0.00 | 0.11; 0.27 |
| Dependent Variable: Hiring Recommendation | | | | |
| Perceived applicant's warmth | 0.38 | 0.06 | 0.00 | 0.26; 0.50 |
| Work setting | −0.31 | 0.18 | 0.08 | −0.66; 0.04 |
| Middle-aged (vs. younger) applicant | 0.13 | 0.39 | 0.75 | −0.63; 0.88 |
| Older (vs. younger) applicant | −0.28 | 0.40 | 0.49 | −1.06; 0.51 |
| Work setting x middle-aged (vs. younger) applicant | 0.12 | 0.25 | 0.62 | −0.37; 0.61 |
| Work setting x older (vs. younger) applicant | −0.05 | 0.25 | 0.83 | −0.54; 0.44 |
| Participants' age | 0.00 | 0.00 | 0.92 | −0.01; 0.01 |
| Participants' nationality | −0.04 | 0.05 | 0.47 | −0.13; 0.06 |
| Participants' gender | −0.13 | 0.10 | 0.19 | −0.33; 0.07 |
| Home-office interference | −0.19 | 0.04 | 0.00 | −0.27; −0.12 |
| Perceived digital skills of the applicant | 0.22 | 0.05 | 0.00 | 0.13; 0.32 |
| Conditional Indirect Effects of Work Setting at Levels of Applicant Age | | | | |
| | Applicant Age | B | SE | 95% CI |
| | Young | 0.05 | 0.06 | −0.07; 0.17 |
| | Middle-aged | 0.02 | 0.06 | −0.09; 0.14 |
| | Old | −0.12 | 0.07 | −0.27; 0.01* |
| Conditional Direct Effects of Work Setting at Levels of Applicant Age | | | | |
| | Applicant Age | B | SE | 95% CI |
| | Young | −0.31 | 0.18 | −0.66; 0.04 |
| | Middle-aged | −0.19 | 0.17 | −0.53; 0.15 |
| | Old | −0.37 | 0.18 | −0.71; −0.02 |

Note. Work setting was coded as follows: 1 = Office, 2 = Teleworking; Applicant age was coded as follows: 1 = Younger, 2 = Middle-aged, 3 = Older; Applicant age was dummy coded with the younger applicant condition as the reference group; CI = confidence interval.

*The 95% confidence interval for older applicants narrowly includes zero. This suggests the effect is approaching significance but does not meet the conventional 95% threshold and should be interpreted with caution.

## Discussion

Study 3 focused exclusively on the interaction between work setting (teleworking versus office) and applicant age (younger, middle-aged, older), omitting applicant gender to allow for a more targeted investigation of age-related effects. Consistent with the findings of Studies 1 and 2, the results supported our hypothesis (Hypothesis 1a) that older applicants are more likely to be recommended for hire when the position involves working from an office rather than teleworking.

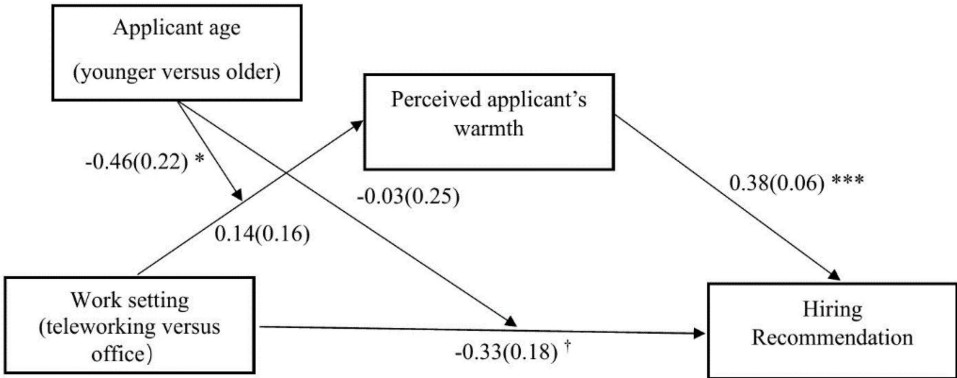

**Fig 6. Work Setting on Hiring Recommendation Through Perceived Applicant's Warmth as a Function of Applicant Age (Study 3).** Note. The path values are the path coefficients with standard errors. † $p < .10$, *$p < .05$, ***$p < .001$.

Importantly, and in line with Hypothesis 1b, this effect was mediated by perceptions of warmth. That is, older applicants were seen as warmer, which in turn increased their likelihood of being hired for an office-based position, though this indirect effect needs to be interpreted with caution, as it approached but did not reach conventional statistical significance. However, the effect of work setting was not significant for younger or middle-aged applicants. Moreover, and consistent with Studies 1 and 2, Study 3 also did not provide evidence supporting the proposed mediating role of competence in explaining the effects of work setting and applicant demographics on hiring recommendations. These findings suggest that perceptions of warmth are particularly salient for older applicants in contexts that emphasize face-to-face collaboration and interpersonal interaction, such as traditional office environments. They also highlight that age-related stereotypes, particularly those involving warmth, can shape recruiter preferences in ways that are contingent on the nature of the work setting.

## General discussion

The overarching aim of this research, across three studies, was to examine how work setting (teleworking versus office) interacts with applicant's demographics, particularly age and gender, in shaping hiring recommendations. Studies 1 and 2 tested interactions between work setting, age, and gender, while Study 3 focused solely on the work setting by age interaction, using a more nuanced three-level age manipulation. Across studies, a consistent pattern emerged: older applicants were more likely to be recommended for hire in office settings compared to telework settings. Study 3 further revealed that this preference was mediated by perceptions of older applicants as warmer. No significant effects emerged for younger or middle-aged applicants, and gender did not interact with work setting in predicting recommendation for hire. Taken together, our findings underscore the presence of age-related biases in hiring preferences, particularly favoring older applicants for office-based over teleworking positions. This pattern suggests that recruiters may perceive older candidates as better suited for environments requiring in-person collaboration and interpersonal engagement yet less suited for remote work environments. Interestingly, no such bias emerged for younger applicants, suggesting that they are perceived as equally suitable for both teleworking and office work settings.

Crucially, no significant gender effects emerged across Studies 1 and 2, suggesting that applicant gender did not influence hiring preferences between teleworking and office settings. This finding aligns with a growing body of literature showing mixed evidence regarding gender discrimination in recruitment. While earlier research has documented clear gender-based disadvantages, particularly for women [46–49], more recent Europe-based studies present a more complex picture: Some experiments report a diminishing or even reversing gender gap, with slight advantages for women in certain

recruitment contexts [50–53]. Moreover, Fernandez-Lozano et al. [54] show that effects related to gender vary significantly depending on job type and context. It is therefore possible that in our study (focusing on job positions like project coordination) gender-based biases were less likely to be activated, particularly when compared to stronger, more salient demographic characteristics such as age.

## Theoretical implications

This research offers several important theoretical implications. First, it extends prior work on selection processes by demonstrating that recruiters' hiring recommendations are not solely shaped by applicants' qualifications but by the interaction between applicants' demographics, specifically age, and the specific requirements of teleworking versus office-based positions. In doing so, it builds on and refines earlier findings [1] by showing that the perceived fit between applicant demographics and work setting plays a critical role in shaping hiring recommendations. Second, it advances the application of Stereotype Content Theory in organizational contexts by demonstrating that age-based stereotypes, particularly perceptions of warmth, can shape hiring decisions depending on the work setting. While older workers are generally perceived as warm but less competent [13,14,16,55], our findings suggest that these warmth perceptions become especially influential in office-based settings, where social interaction and collaboration are more prominent. This supports the idea that contextual cues (e.g., work setting or organizational environment) activate different stereotype dimensions and influence decision-making. Additionally, the mediating role of warmth highlights the relevance of social perception processes in hiring biases, extending prior work that has predominantly focused on competence-based evaluations [1,31]. Finally, the absence of consistent gender effects across studies underscores the importance of distinguishing between demographic categories when studying bias, and suggests that age may be a more salient or stable basis for stereotype-driven hiring decisions in the context of evolving work arrangements like telework versus office [50–53].

## Practical implications

The findings of this research also have important practical implications for organizations aiming to foster fair and inclusive hiring practices in an era of hybrid and remote work. First, our results suggest that hiring decisions can be influenced by subtle age-related stereotypes, especially in relation to perceived fit between applicant age and work setting. Organizations should therefore invest in bias-awareness training for recruiters, emphasizing how assumptions about age-related traits such as warmth or competence may unconsciously influence their evaluations. Second, standardized hiring procedures should be adopted to minimize the role of subjective perceptions, particularly in contexts where telework or office presence is a factor. Third, job postings and hiring criteria should be carefully worded to focus on job or role-relevant competencies rather than implicit social expectations tied to work settings. Lastly, these insights are especially relevant in light of the ongoing digital transformation and flexible work arrangements, urging HR departments to ensure that evolving workplace structures do not inadvertently disadvantage certain demographic groups, especially older job applicants.

## Strengths and limitations

A key strength of this research lies in its multi-study design, which systematically examined how applicant age and work setting interact to influence hiring recommendations. By conducting three experimental studies and manipulating applicant age across different ranges (including a refined three-level design in Study 3), the research offers a more nuanced understanding of age-related hiring biases. Furthermore, the inclusion of warmth (and competence) as potential mediators provides insight into the psychological mechanisms underpinning age-related preferences, building on Stereotype Content Theory in an applied organizational context. The involvement of professional recruiters in Study 3 is an additional strength, as it increases the ecological validity of the findings and suggests that the observed patterns may reflect decision-making processes in real-world selection contexts. Finally, the vignette used across studies 2 and 3 (describing a vacancy for a

project coordinator role) was pilot tested to ensure appropriateness for the experimental manipulations, further strengthening the methodological rigor of the research.

Despite its strengths, the research has several limitations that should be acknowledged. First, although the experimental manipulations were successful, the use of hypothetical applicant profiles may not fully capture the complexity of real-life recruitment decisions, where factors such as organizational culture, recruiter characteristics, and applicant interaction can further influence evaluations. Second, the non-significant gender effects observed across Studies 1 and 2 may be partly context-dependent and could vary across industries or job types. These non-significant effects might also reflect broader shifts in gender norms or the specific characteristics of the job role used in the vignettes (project manager and project coordinator). Third, the involvement of non-professional recruiters in Studies 1 and 2 (and partly in Study 3) may have influenced the pattern of findings, particularly the absence of significant interaction effects involving gender, or between work setting and age in predicting perceived applicant warmth and competence (mediators). In other words, participants' limited experience with recruitment decisions may have led to more generalized or less differentiated evaluations, compared to professional recruiters in Study 3. Fourth, the composition of our samples in Studies 1 and 2 warrants consideration as the majority of participants were women. The predominance of female participants in these studies may raise concerns about generalizability, as well as the possibility of homophily effects (e.g., participants favoring applicants who share their own demographic characteristics). Additionally, our vignette design asked participants to evaluate applicants in isolation, whereas real hiring contexts typically involve comparing multiple candidates simultaneously. While this design strengthens internal validity by allowing us to draw causal inferences about the effects of work setting and demographics, it remains unclear whether the same patterns would hold in competitive applicant pools. Crucially, our hiring recommendation measure was treated as a continuous outcome, which is consistent with prior research [1,31] but may not fully capture qualitative thresholds in decision making (e.g., recommend vs. not recommend). Future research should examine whether our findings hold when using dichotomous or categorical outcomes that more directly reflect final hiring choices. Furthermore, our recruitment largely relied on convenience sampling rather than random sampling from a population of recruiters. This introduces the possibility of selection bias, as participants who volunteered may differ systematically from the broader population --for example, in terms of age and gender distribution, or interest in organizational topics. While this does not compromise the internal validity of our experimental design, it does limit the external validity of our findings. Moreover, the present studies were -primarily- conducted in the Netherlands, which has its own cultural and institutional characteristics (e.g., relatively strong labor protections, widespread acceptance of flexible work arrangements). These features may limit the generalizability of our findings to other settings or countries with different labor market structures and cultural norms.

Finally, a further limitation concerns the high intercorrelation between warmth and competence in our data. When both competence and warmth were included in the same model, the effects attenuated, likely due to the high correlation between the two constructs, rather than the absence of an effect. This issue is not unique to our study: warmth and competence are well-established as distinct yet strongly related dimensions of social judgment which can be difficult to disentangle empirically [56]. In line with prior research using Stereotype Content Theory [39], we therefore examined them separately, while acknowledging that this approach does not allow us to determine their unique effects over and above one another. We should note that the strong positive correlation we observed between perceived warmth and competence has theoretical implications too: It indicates that these dimensions do not function as opposing stereotypes, as is sometimes emphasized in compensation accounts (see compensation effect; [57]. Instead, evaluators may rely on a halo effect [58], such that positive evaluations on one dimension (e.g., warmth) spilled over into positive judgments on the other (competence). This interpretation is consistent with prior work showing that warmth and competence, while theoretically distinct, are often positively related in practice [39,56]. Although a full exploration of this issue (i.e., comparing compensation with halo effect accounts) lies beyond the scope of the present study, our results indicate that warmth perceptions

were particularly decisive in explaining preferences for older applicants in office settings, whereas competence perceptions did not emerge as significant mediators.

## Future directions

Future research should investigate intersectional effects of age, gender, and other demographics (e.g., race/ethnicity or parental status) in hiring decisions across a broader range of occupations. Intersectional biases can manifest uniquely, with individuals belonging to multiple marginalized groups potentially facing compounded disadvantages [7]. For instance, [Wilson et al., 59] demonstrated that resume screening exhibits significant biases against Black male applicants, emphasizing the need to scrutinize recruitment processes. Similarly, Erlandsson [9] found that ethnic discrimination in hiring persists regardless of recruiter gender. These findings suggest that biases may be influenced by the interplay of various demographic characteristics, necessitating studies that consider these intersections.

Moreover, employing field experiments and involving actual recruiters can enhance the ecological validity of future studies. Real-world settings allow for the observation of actual decision-making processes, capturing nuances that laboratory experiments may overlook. Additionally, investigating organizational interventions aimed at mitigating age-based biases is crucial. Structured interviews, for example, have been shown to reduce bias by standardizing the evaluation process, ensuring that all candidates are assessed based on the same criteria [60]. Diversity training programs, when designed effectively, can also raise awareness of unconscious biases and promote inclusive hiring practices [61,62]. Exploring the efficacy of these interventions in evolving work contexts will offer important insights into creating equitable hiring practices.

## Conclusions

In conclusion, the present research shows that hiring recommendation is influenced by the interaction between work setting and applicant age. Across studies, older applicants were more likely to be recommended for an office-based position than for a telework position (all three studies), and this effect was mediated by the applicant's perceptions of warmth (Study 3). Interestingly, no such pattern emerged for younger or middle-aged applicants. This suggests that age-based biases are more pronounced for older individuals in contexts emphasizing social interactions and face-to-face contact. Additionally, applicant gender did not significantly influence hiring decisions in either work setting, pointing to age as the more salient demographic in shaping recruiters' judgments in in-office work environments [see also 63].

## Supporting information

**S1 File. S1_Inclusivity_Questionnaire.**
(DOCX)

**S2 File. S2_Online supplemental material.**
(DOCX)

## Acknowledgments

Declaration of generative AI and AI-assisted technologies in the writing process. During the preparation of this work the authors used ChatGPT in order to improve the language of this paper. After using this tool, the authors reviewed and edited the content as needed and take full responsibility for the content of the publication. No AI was used to generate or create any content of the paper.

## Author contributions

**Conceptualization:** Kyriaki Fousiani, Bibiana M. Armenta Gutierrez.

**Data curation:** Kyriaki Fousiani, Sylvia Xu.

**Formal analysis:** Kyriaki Fousiani, Sylvia Xu.

**Investigation:** Kyriaki Fousiani, Bibiana M. Armenta Gutierrez, Chloe Sypes.

**Methodology:** Kyriaki Fousiani, Sylvia Xu, Bibiana M. Armenta Gutierrez, Chloe Sypes.

**Project administration:** Kyriaki Fousiani.

**Resources:** Kyriaki Fousiani.

**Software:** Kyriaki Fousiani, Bibiana M. Armenta Gutierrez.

**Supervision:** Kyriaki Fousiani.

**Validation:** Kyriaki Fousiani.

**Writing – original draft:** Kyriaki Fousiani, Sylvia Xu.

**Writing – review & editing:** Kyriaki Fousiani, Bibiana M. Armenta Gutierrez, Sylvia Xu.

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
