## [Decision Letter · Decision Letter 0]

30 Sep 2025

PONE-D-25-34010Too Old to Telework? Age but not Gender Shapes Hiring Biases Across Telework and Office SettingsPLOS ONE?

Dear Dr. Fousiani,

We look forward to receiving your revised manuscript.

Kind regards,

Mattia Vacchiano, Ph.D.

Academic Editor

PLOS ONE

Journal Requirements:

“Authors declare that they have no conflict of interest.”

4. Thank you for uploading your study's underlying data set. Unfortunately, the repository you have noted in your Data Availability statement does not qualify as an acceptable data repository according to PLOS's standards.

Reviewers' comments:

Reviewer's Responses to Questions

**Comments to the Author**

1. Is the manuscript technically sound, and do the data support the conclusions?

Reviewer #1: Partly

Reviewer #2: Yes

2. Has the statistical analysis been performed appropriately and rigorously?

Reviewer #1: Yes

Reviewer #2: Yes

3. Have the authors made all data underlying the findings in their manuscript fully available?

Reviewer #1: Yes

Reviewer #2: Yes

4. Is the manuscript presented in an intelligible fashion and written in standard English?

Reviewer #1: Yes

Reviewer #2: Yes

Reviewer #1: Thanks for this insightful and fine-grained paper, and for the contribution it makes to existing literature. Here I outline a series of questions and points that I’d like to ask you about:

- “Considering the strong positive correlation between warmth and competence (see Table 1) we conducted separate analyses for each mediator in order to identify their unique contribution to the model. » This may introduce an omitted variable bias in the regressions. What is the behavior of the model when both are introduced? What’s the obtained VIF for these variables? Please provide more details on this or discuss potential drawbacks for the analysis (applies to Study 3 as well).

- On a conceptual side, this positive correlation could have implications for the theoretical framing of the paper and existing literature, where gender/age are presented as if they were evaluated in warmth/competence as mutually exclusive, or in an opposed way.

- Figure 1: The theory seems to argue that the work setting has an effect on hiring probabilities by gender and/or by age, due to different pereceived characteristics of these groups. Therefore, women would be perceived as having more warmth, and younger workers as being more competent. In empirical tests, however, the interaction between characteristics x work setting implies that work setting changes the perception of the different groups. Could you please clarify this point.

- Is there are specific reason why the main regression tables of studies 1 and 2 are not reported? As they provide more details than just the predicted hiring recommendation reported in figures.

- Study 3: are there differences between professional recruiters and students in the way they evaluate profiles and formulate hiring recommendations? Is there intergroup consistency? Can these be compared so they can potentially shed light on the interpretation of Studies 1 and 2, where there were no professional recruiters?

- Study 3: even if applicant gender was not manipulated in this study, gender-specific behavior of participants regarding the evaluation of profiles may still emerge (as nationality is not varied in the treatment, yet it’s controlled for). Please provide clarification on this decision.

General discussion:

- Given that warmth only plays a role in Study 3 for old-age applicants, shouldn't the affirmation "These findings suggest that hiring decisions are shaped not just by applicant qualifications, but also by perceived fit between applicant demographics and contextual demands of the job." perhaps be framed in a more qualified way in the abstract?

- “Crucially, no significant gender effects emerged across the three studies, suggesting that applicant gender did not influence hiring preferences in either teleworking or office settings. » Rather, wouldn’t it be between teleworking and office settings? Results concerning absolute hiring probabilities are not examined in detailed manner.

- What’s the evidence that recruited participants behave in ways that are comparable to actual recruiters? Please provide some clarification, based on similar study designs – or the two groups of participants of Study 3.

- What are the practical implications real life processes where a) applicants shape competition with other profiles, rather than being evaluated by themselves and b) hiring decisions are dichotomic as yes/no? Is a linear increase in recommendation probability informative if -on a scale from 1 to 7- it goes from 1 to 2, as it would be if it goes from 3 to 4? (as this could potentially tilt the balance on a final decision). Have sensitivity tests of this sort been conducted (as in more likely to recommend v. not likely to recommend)? In case they’re not considered pertinent, why would it be the case?

Reviewer #2: The topic is highly interesting, and overall, the studies are well designed with compelling arguments. However, there are several recommendations that would strengthen the methodological rigor, enhance transparency, and help avoid potential overclaiming of the results.

First, I suggest including in an appendix the full instructions shown to participants, along with all the questions used in the study. Providing these materials would improve replicability and allow readers to better evaluate the validity of the research design.

Second, the paper should report descriptive statistics of the sample (e.g., age, gender distribution, educational background, prior experience with the topic). This information is essential for transparency and for assessing the extent to which the findings can be generalized.

Third, an important limitation that must be acknowledged is the composition of the sample. The majority of participants are women and very young individuals, which may introduce bias. Given the likelihood of homophily effects—such as young participants showing preference for peers—this imbalance should be explicitly discussed.

Fourth, the recruitment process was not random, and this limitation needs to be acknowledged and explored. A more detailed discussion of potential selection bias and how it might have influenced the findings would strengthen the paper’s credibility.

Finally, I recommend that the authors explicitly address both the limitations and advantages of vignette studies. While they provide a controlled and cost-efficient way to study decision-making, their artificial nature can affect external validity. Similarly, the discussion should consider the specific characteristics of the Dutch context in which the research was conducted, as these may limit generalizability to other cultural or institutional settings.

**Do you want your identity to be public for this peer review?** For information about this choice, including consent withdrawal, please see our Privacy Policy

Reviewer #1: No

Reviewer #2: No

---

## [Author Response · Author response to Decision Letter 1]

7 Nov 2025

Response to Reviewers

We would like to sincerely thank both Reviewers for their thorough and insightful evaluations of our manuscript. We greatly appreciate the time and effort they dedicated to providing constructive feedback and valuable suggestions. Below, we offer a detailed, point-by-point response outlining how we have addressed each of their comments and concerns.

Reviewer #1

Thanks for this insightful and fine-grained paper, and for the contribution it makes to existing literature. Here I outline a series of questions and points that I’d like to ask you about.

Authors: We sincerely thank you for the time and effort you dedicated to reviewing our paper and for your insightful comments, which have helped us to further strengthen the manuscript.

Reviewer 1 -point 1: “Considering the strong positive correlation between warmth and competence (see Table 1) we conducted separate analyses for each mediator in order to identify their unique contribution to the model. » This may introduce an omitted variable bias in the regressions. What is the behavior of the model when both are introduced? What’s the obtained VIF for these variables? Please provide more details on this or discuss potential drawbacks for the analysis (applies to Study 3 as well).

Authors: We thank the reviewer for this thoughtful observation. Following the suggestion, we re-ran the mediation analyses including both warmth and competence simultaneously as mediators. When both variables were entered in the same model, the indirect effects were no longer statistically significant. This attenuation likely reflects the high correlations between warmth and competence, which has been repeatedly documented in the literature (e.g., Fiske et al., 2002; Judd et al., 2005).

To further examine whether multicollinearity could explain these results, we conducted diagnostic tests based on the reviewer’s recommendation. The Variance Inflation Factors (VIFs) for warmth and competence were all below the commonly accepted threshold (VIF < 5), indicating that multicollinearity was not statistically problematic.

Nevertheless, given the strong correlation between warmth and competence and their interdependent nature in social perception, we consider it more conceptually and empirically appropriate to analyze each mediator separately. This approach is consistent with prior studies examining similar constructs (e.g., Abele & Wojciszke, 2014; Judd et al., 2005), which typically test the mediating roles of warmth and competence in separate models to avoid mutual suppression and redundancy effects that can obscure interpretation.

In sum, although our collinearity diagnostics did not reveal statistical concerns, we decided to retain the more conservative analytical strategy (testing warmth and competence as separate mediators) to ensure conceptual clarity, comparability with prior work, and robustness of interpretation. Moreover, we clarify the behavior of the combined model on p.2 (“When warmth and competence were entered as simultaneous mediators, their effects on hiring recommendation were no longer significant. This is likely due to their strong correlation and their interdependent nature in social perception. Therefore, consistent with prior research (e.g., Fiske et al., 2002; Abele & Wojciszke, 2014), we examined them separately to avoid redundancy and ensure conceptual clarity”). We also acknowledge the limitation that our analytic strategy does not allow us to determine unique effects over and above one another. We also note that this correlation itself is informative, as it reflects the challenge of disentangling warmth and competence. Finally, we refer to this limitation in the General Discussion, p. 30 (“Finally, a further limitation concerns the high intercorrelation between warmth and competence in our data. When both competence and warmth were included in the same model, the effects attenuated, likely due to their empirical overlap, rather than the absence of an effect. This issue is not unique to our study: warmth and competence are well-established as distinct yet strongly related dimensions of social judgment which can be difficult to disentangle empirically (Judd et al., 2005). In line with prior research using Stereotype Content Theory (Abele & Wojciszke, 2014), we therefore examined them separately, while acknowledging that this approach does not allow us to determine their unique effects over and above one another”).

We further acknowledge the theoretical implications of the strong correlation between the two constructs. Please see our response to your second point regarding this issue.

Reviewer 1 -point 2: On a conceptual side, this positive correlation could have implications for the theoretical framing of the paper and existing literature, where gender/age are presented as if they were evaluated in warmth/competence as mutually exclusive, or in an opposed way.

Authors: We appreciate this thoughtful observation. In revising the manuscript, we clarified that our intention was not to treat warmth and competence as mutually exclusive dimensions. Rather, we noted that the strong positive correlation between the two in our data suggests they did not function as opposing stereotypes, as sometimes emphasized in compensation accounts (Kervyn, Yzerbyt, & Judd, 2010). Instead, evaluators may have relied on a halo effect (Thorndike, 1920), whereby positive evaluations on one dimension spilled over into the other. This interpretation is consistent with prior work showing that warmth and competence, while conceptually distinct, are often positively related in practice (Judd et al., 2005; Abele & Wojciszke, 2014). We have added a discussion of these conceptual implications (p. 30). Importantly, our results still revealed that warmth perceptions were particularly decisive in explaining preferences for older applicants in office settings, whereas competence did not emerge as a significant mediator. We believe this addition strengthens the theoretical contribution of the paper by highlighting how contextual demands may determine which dimension is more salient in recruitment decisions, even when halo effects blur the boundaries between them. To further address your comment in the paper, in the General Discussion (p. 30) we have now added this paragraph: “We should note that the strong positive correlation we observed between perceived warmth and competence has theoretical implications too: It indicates that these dimensions do not function as opposing stereotypes, as is sometimes emphasized in compensation accounts (see compensation effect; Kervyn et al., 2010). Instead, evaluators may rely on a halo effect (Thorndike, 1920), such that positive evaluations on one dimension (e.g., warmth) spilled over into positive judgments on the other (competence). This interpretation is consistent with prior work showing that warmth and competence, while theoretically distinct, are often positively related in practice (Judd et al., 2005; Abele & Wojciszke, 2014). Although a full exploration of this issue (i.e., comparing compensation with halo effect accounts) lies beyond the scope of the present study, our results indicate that warmth perceptions were particularly decisive in explaining preferences for older applicants in office settings, whereas competence perceptions did not emerge as significant mediators”.

Reviewer 1 -point 3: Figure 1: The theory seems to argue that the work setting has an effect on hiring probabilities by gender and/or by age, due to different pereceived characteristics of these groups. Therefore, women would be perceived as having more warmth, and younger workers as being more competent. In empirical tests, however, the interaction between characteristics x work setting implies that work setting changes the perception of the different groups. Could you please clarify this point.

Authors: We appreciate this insightful comment. We would like to use this opportunity to clarify that our theoretical focus is not on the main effects of work setting, age or gender per se, but on how work setting interacts with these demographic characteristics to shape hiring outcomes through warmth and competence. Indeed, as you pointed out, women and older people are typically perceived as warmer, whereas men and younger people are often perceived as more competent. However, these main effects of gender and age on warmth and competence have already been well established in prior research and are summarized in our theory section. Our contribution lies in examining when and how work setting—also previously shown to influence warmth- and competence-related perceptions (Wojciszke & Abele, 2008; Fousiani et al., 2022)—becomes particularly consequential depending on the applicant’s demographic characteristics. In other words, we argue that work setting influences perceptions of applicant warmth and competence, which in turn guide hiring recommendations, but the strength of these effects differs for older versus younger applicants and for female versus male applicants. Thus, in our model, demographics act as boundary conditions for the impact of work setting, rather than as independent drivers of stereotype content. This approach allows us to extend existing theory by showing how established stereotypes (about age or gender) influence hiring decisions across contemporary work contexts such as telework versus office settings.

Of course, an alternative model could position demographics as the independent variables influencing hiring decisions through warmth and competence, with work setting serving as the moderator. While this represents an interesting angle, our study places the emphasis on work setting as the core variable (independent variable), given its growing relevance in shaping applicant perceptions and hiring decisions in contemporary workplaces.

Reviewer 1 -point 4: Is there are specific reason why the main regression tables of studies 1 and 2 are not reported? As they provide more details than just the predicted hiring recommendation reported in figures.

Authors:

We thank you for noticing this omission. The absence of the main regression tables for Studies 1 and 2 was unintentional. We have now included these tables in the revised manuscript (please refer to Tables 2 to 5), which report the full regression results underlying the predicted hiring recommendations shown in the figures.

Reviewer 1 -point 5: Study 3: are there differences between professional recruiters and students in the way they evaluate profiles and formulate hiring recommendations? Is there intergroup consistency? Can these be compared so they can potentially shed light on the interpretation of Studies 1 and 2, where there were no professional recruiters?

Authors:

We thank you for this valuable suggestion regarding subgroup analyses. We chose not to conduct analyses separately for recruiters (N = 185) and non-recruiters (N = 143) because the subgroup sample sizes would be underpowered to detect the expected interaction effects. Sensitivity analyses using G*Power indicate that with N = 143, the minimum detectable effect size (f) at 80% power and α = 0.05 is 0.26, and with N = 185, it is 0.23. Based on Studies 1 and 2, the expected effect would be f ≈ 0.13–0.15, which is substantially smaller than the detectable effects in either subgroup. Consequently, subgroup analyses would have a high risk of Type II error and could yield unreliable results. By analyzing the full sample (N = 328), we achieved sufficient power to detect small-to-medium effects (minimum detectable f = 0.17), making the results more robust and reliable.

However, we do recognize this as a limitation in our research and as an important direction for future research, as suggested in your other comments as well.

Reviewer 1 -point 6: Study 3: even if applicant gender was not manipulated in this study, gender-specific behavior of participants regarding the evaluation of profiles may still emerge (as nationality is not varied in the treatment, yet it’s controlled for). Please provide clarification on this decision.

Authors: We thank the reviewer for raising this important point. Although applicant gender was not manipulated in Study 3, we agree that participants’ own gender could potentially shape their evaluation of profiles. To address this concern, we have now included participants’ gender as an additional control variable in all analyses for Study 3 and reported the updated results in the revised manuscript (Pages 15, 17-24). The inclusion of gender did not meaningfully change the results—the direction, strength, and significance of the main effects remained consistent—indicating that our findings are robust and not driven by gender-specific evaluation patterns.

General discussion:

Reviewer 1 -point 7: Given that warmth only plays a role in Study 3 for old-age applicants, shouldn't the affirmation "These findings suggest that hiring decisions are shaped not just by applicant qualifications, but also by perceived fit between applicant demographics and contextual demands of the job." perhaps be framed in a more qualified way in the abstract?

Authors: We appreciate this observation. We would like to clarify that in the sentence highlighted, we did not intend to suggest that warmth universally mediated effects across all studies. Instead, the sentence refers more broadly to our main finding: hiring decisions were consistently shaped by the interaction between applicant demographics (in this case, age) and work setting. “These findings suggest that hiring decisions are shaped not just by applicant qualifications, but also by perceived fit between applicant (age-related) demographics and contextual demands of the job”. Naturally, we remain open to further suggestions should the current version still risk being read as misrepresenting our results.

Reviewer 1 -point 8: “Crucially, no significant gender effects emerged across the three studies, suggesting that applicant gender did not influence hiring preferences in either teleworking or office settings. » Rather, wouldn’t it be between teleworking and office settings? Results concerning absolute hiring probabilities are not examined in detailed manner.

Authors: Thank you for pointing out this inconsistency. You are right and we have made the change suggested in your comment.

Reviewer 1 -point 9: What’s the evidence that recruited participants behave in ways that are comparable to actual recruiters? Please provide some clarification, based on similar study designs – or the two groups of participants of Study 3.

Authors: Thank you for raising this important issue. Previous research has indeed employed similar methodological approaches and participant samples to study hiring intentions and recruitment recommendations. For example, Fousiani, Van Prooijen, and Armenta (2022) and Fousiani, Sypes, and Armenta (2023) used multi-study designs in which participants either consisted of students and employees from various organizations asked to assume the role of recruiters, or actual recruiters. Crucially, their findings replicated across these different samples, suggesting that participants evaluating applicants in an experimental setting behave in ways that are comparable to actual recruiters. This evidence supports the validity of our approach, in which participants assessed applicants presented in vignette-based hiring scenarios.

Reviewer 1 -point 10: What are the practical implications real life processes where a) applicants shape competition with other profiles, rather than being evaluated by themselves and b) hiring decisions are dichotomic as yes/no?

Authors:

We thank the reviewer for this insightful comment. We agree that our design simplifies the complexity of real-world hiring. In actual recruitment, applicants compete with each other in relative comparison, and decisions are typically dichotomous (yes/no) rather than expressed on a continuous rating scale. We recognize that this design does not fully capture competitive dynamics or the binary nature of final hiring outcomes. Yet, our approach, in line with prior recruitment research, allows us to isolate the effects of applicant demographics and work setting on hiring intentions under controlled conditions, without the confounding influence o

---

## [Decision Letter · Decision Letter 1]

21 Dec 2025

Too old to telework? Age but not gender shapes hiring biases across telework and office settings

PONE-D-25-34010R1

Dear Dr. Kyriaki Fousiani,

We’re pleased to inform you that your manuscript has been judged scientifically suitable for publication and will be formally accepted for publication once it meets all outstanding technical requirements.

Kind regards,

Mattia Vacchiano, Ph.D.

Academic Editor

PLOS One

Reviewers' comments:

Reviewer's Responses to Questions

**Comments to the Author**

Reviewer #1: All comments have been addressed

Reviewer #2: All comments have been addressed

2. Is the manuscript technically sound, and do the data support the conclusions?

Reviewer #1: Yes

Reviewer #2: Yes

3. Has the statistical analysis been performed appropriately and rigorously?

Reviewer #1: Yes

Reviewer #2: Yes

4. Have the authors made all data underlying the findings in their manuscript fully available?

Reviewer #1: Yes

Reviewer #2: No

5. Is the manuscript presented in an intelligible fashion and written in standard English?

Reviewer #1: Yes

Reviewer #2: Yes

Reviewer #1: (No Response)

Reviewer #2: The paper is interesting and offers a meaningful and original contribution. It is clearly written, conceptually coherent, and addresses an underexplored and timely dimension in a way that advances the field.

In my view, it deserves to be published.

**Do you want your identity to be public for this peer review?** For information about this choice, including consent withdrawal, please see our Privacy Policy

Reviewer #1: No

Reviewer #2: No

---

## [Editor Report · Acceptance letter]

PONE-D-25-34010R1

PLOS One

Dear Dr. Fousiani,

I'm pleased to inform you that your manuscript has been deemed suitable for publication in PLOS One. Congratulations! Your manuscript is now being handed over to our production team.

Kind regards,

on behalf of

Dr. Mattia Vacchiano

Academic Editor

PLOS One